



# The effect of rainfall amount and timing on annual transpiration in grazed savanna grassland

Matti Räsänen[1], Mika Aurela[2], Ville Vakkari[2], Johan P. Beukes[3], Juha-Pekka Tuovinen[2], Pieter G. Van Zyl[3], Miroslav Josipovic[3], Stefan J. Siebert[3], Tuomas Laurila[2], Markku Kulmala[1], Lauri Laakso[2,3], Janne Rinne[4], Ram Oren[5,6] and Gabriel Katul[5,7]

[1]Institute for Atmospheric and Earth System Research, University of Helsinki, Finland
[2]Finnish Meteorological Institute, Helsinki, Finland
[3]Unit for Environmental Sciences and Management, North-West University, South Africa
[4]Department of Physical Geography and Ecosystem Science, Lund University, Sweden
[5]Nicholas School of the Environment, Duke University, Durham, North Carolina, USA
[6]Department of Forest Science, University of Helsinki, Finland
[7]Department of Civil and Environmental Engineering, Duke University, Durham, North Carolina, USA
*Correspondence to*: Matti Räsänen (matti.rasanen@helsinki.fi)

## Abstract

The role of precipitation (P) variability on evapotranspiration (ET) and its two components transpiration (T) and evaporation (E) rates from savannas continues to draw significant research interest given its relevance to a number of eco-hydrological applications. The work here reports on six years of measured ET and energy flux components, and estimated T from a grazed savanna grassland collected at a research site situated in Welgegund, South Africa. During this period, annual P varied considerably in amount (421 mm to 614 mm), rainy season length and precipitation intensity. T was estimated using annual water use efficiency and gross primary production determined from eddy-covariance measurements of net ecosystem $CO_2$ exchange rates. The computed annual T was highly constrained to $352 \pm 8$ mm (T/ET=0.55) for four wet years when rainfall was near or above the long-term mean. This is explained by the near constant annual tree transpiration and moderate water stress of $C_4$ grasses during these years. In a drought year with intermittent rainfall, the annual ecosystem T was reduced due to grass dieback-regrowth that alters the temporal dynamics of bare soil cover and infiltration, and complicates monthly T/ET relation to leaf-area index (LAI). However, annual ET remains approximately equal to annual precipitation (P) even during the drought year due to increased soil evaporation. Indeed, at annual scales, ET $\approx$ P and annual T is conservative despite variation in amount and timing in rainfall, due to constant water use of mature trees, and the ability of $C_4$ grasses to maintain transpiration at moderate water stress and effectively use pulsed rainfall.



# 1 Introduction

Wooded grasslands provide an important ecosystem service in the form of grazing in South Africa. In this semi-arid zone, almost all the annual precipitation ($P \approx 500$ mm) is consumed by evapotranspiration (ET) (ET/P is ~ 1.0), lost as transpiration (T) and evaporation (E) of similar magnitude (Zhang et al., 2001). The T component evaporates from stomata of leaves of the

sparse tree component (~15 % of the ground cover), seasonal grasses, and the small amount of forbs. The E component is high following rain events but can persist as a loss from the rooting zone to the atmosphere over extended periods (> week) as discussed elsewhere (Perez-Priego et al., 2018). The partition of ET between E and T may affect the net radiation and surface temperature on short time scales (sub-daily). However, processes that increase the proportional (and amount of) water used in transpiration, thus facilitating greater carbon uptake and subsequent fodder production for cattle grazing, occur over time scales

of weeks or longer. Because of the link between T and carbon uptake from the atmosphere, there is growing interest in how ET is partitioned into E and T in such semi-arid ecosystems (Merbold et al., 2009; Sankaran et al., 2004; Scanlon et al., 2002, 2005; Scholes and Archer, 1997; Volder et al., 2013; Williams and Albertson, 2004; Xu et al., 2015; Yu and D'Odorico, 2015). The aim of this study is to explore this partitioning using a long-term record of measured fluxes of energy, water, $CO_2$, and vegetation activity. The focus is restricted to processes operating at daily, monthly, and seasonal time scales, commensurate

with controls over the annual partitioning of P into T, and the resulting carbon uptake in gross primary production (GPP).

The two vegetation components of wooded grasslands have distinct seasonal dynamics of leaf area display and physiological activity. In the study site, the main woody species is *Acacia erioloba*, a deep rooted semi-deciduous tree with a low leaf turnover rate, resulting in minor leaf area changes. Furthermore, this species has been shown to absorb 37 % of its water below 1 m depth, decoupling its physiological activity from recent precipitation and shallow soil water content (Beyer et al., 2018).

The perennial $C_4$ grass species at the site have shallower root system and are more physiologically responsive to intermittent rainfall events (Sankaran, 2019). Compared to the $C_3$ trees, the $C_4$ grass layer has $CO_2$ concentrated in the bundle sheath that enables greater efficiency of light and water use in hot and open environment (Ripley et al., 2010). In general, for the same volume of soil, the $C_4$ grass is an intensive and fast user of the soil water, reflected in higher photosynthesis yet not necessarily high gas-exchange sensitivity to water stress (Taylor et al., 2014). However, due to low water storage capabilities of grasses,

severe drought may alter its leaf area dynamics. The objective here is therefore not only to partition measured ET into T and E, but also to quantify the effect of environmental variables on the seasonality of the grass activity.

A number of methods have already been proposed for partitioning ET into T and E using simultaneous eddy covariance measured ET and net ecosystem exchange (NEE) of carbon dioxide. All these methods are premised on the fact that GPP, which can be inferred from eddy-covariance measured NEE and night-time respiration extrapolated to daytime, is linked

through stomatal exchange of $CO_2$ for water vapor to only the T component of ET. From definitions, it directly follows that the flux-based water use efficiency WUE $= (GPP/T) \propto (c_a/VPD)(1 - c_i/c_a)$, where $c_i$ and $c_a$ are the intercellular and ambient atmospheric $CO_2$ concentration, and VPD is the vapor pressure deficit. Based on stomatal optimization theories that maximize carbon gain for a given amount of water in the rooting system per unit leaf area, $(1 - c_i/c_a) \propto \sqrt{VPD}$ as



demonstrated in a number of studies reviewed elsewhere (Hari et al., 2000; Katul et al., 2009, 2010). When these theories are combined with the definition of WUE, $T \propto GPP \times VPD^{0.5}$, provided that $c_a$ does not vary appreciably. The proportionality constant in the aforementioned expression is linked to the marginal water use efficiency (or the Lagrange multiplier in optimal stomatal theories) and must be externally supplied or determined from conditions when $T \approx$ measured ET. When this

proportionality constant is known, eddy-covariance measured GPP (and VPD) can be used to infer T and, as residual, E(=ET-T).

Three different methods that establish relations between T and GPP are used to estimate monthly T/ET. The first, which we call the Berkelhammer et al. (2016) method empirically establishes a T=ET by fitting a line to the minimum ET values, presumed to represent T only (5th percentile), and $GPP \times VPD^{0.5}$ bins. The T=ET line is used to estimate T/ET for each

measured half-hour value. This T/ET estimate was shown to be in agreement with isotopic T/ET measurements (Berkelhammer et al., 2016). The second, the uWUE method uses quantile regression with zero-intercept to fit T=ET line (Zhou et al., 2016). These methods assume that T=ET half-hour values exist in the data and can be determined from minimum ET runs. Also, the optimal response of T to VPD must be assumed, a reasonable assumption based on a recent review (Stoy et al., 2019). The third, the transpiration estimation algorithm (TEA) method, is a nonparametric random forest model that does not assume

optimal response of stomata to VPD but must assume that T/ET approaches 1.0. These methods were chosen for this study because previous application showed some success when applied to multi-site data sets, and thus their uncertainties have been explored at other sites. Comparing these methods here allows selecting the most suitable partitioning scheme for water limited ecosystems.

The overarching question to be addressed here is how ET and its two components (E and T) vary with mean annual P and

vegetation changes in a grazed savanna grassland ecosystem. This work offers a methodological test as to whether such partitioning can be achieved for water-limited ecosystem with fast changes in vegetation cover. The MODIS LAI and EVI were used to quantify the vegetation dynamics at the site. To guide the discussion, three study objectives are formulated: (i) quantifying the variation in P, ET and the estimated T based on meteorological measurements at this new long-term monitoring site, (ii) identifying the main drivers of the annual, seasonal and monthly variation in the water balance components, and (iii)

identifying remote sensing variables that explain variations in T/ET and transpiration.

## 2 Materials and methods

### 2.1 Site description

The Welgegund measurement site is located in a grazed savanna grassland in South Africa (26 ° 34' 10" S, 26 ° 56' 21" E, 1480 m a.s.l.). The research site is a part of a large-scale commercial farm that houses about 1300 cattle head. This number of

cattle head varies by some ± 300 depending on the year. The cattle grazing area is approximately 6000 ha. The area experiences two seasonal periods: a warm rainy season from October to April, and a cool dry season from May to September. The 16-year mean annual rainfall determined at a nearby weather station in Potchefstroom was 540 mm, with a standard





deviation of 112 mm (Räsänen et al., 2017). The soil around the site is loamy sand. The water table depth is not known but the farm well has a water table depth of 30 m with continuous supply of water. The vegetation in the area is an open thornveld. The dominant perennial $C_4$ grass species are *Eragrostis trichophora*, *Panicum maximum* and *Setaria sphacelata*. The tree cover is 15 % and the dominant tree species is *Vachellia erioloba* and other less prominent species such as *Celtis africana* and

*Searsia pyroides*. There are also forbs, of which the dominant species are *Dicoma tomentosa*, *Hermannia depressa*, *Pentzia globosa* and *Selago densiflora*. More details about the site and vegetation cover may be found elsewhere (Jaars et al., 2016; Räsänen et al., 2017).

## 2.2  Measurements

At the research site, atmospheric aerosols, trace gases and meteorological variables are measured continuously. Here we

describe the measurements directly related to energy and water balance. The eddy covariance system consists of a triaxial sonic anemometer (METEK USA-1) and a Li-Cor (LI-7000) closed path infrared gas analyzer, both positioned at 9 m above the ground surface. The sampling frequency of the eddy covariance system was 10 Hz. The gas analyzer was calibrated every month with a high-precision $CO_2$ span gas using synthetic air with $CO_2 < 0.5$ ppm as a reference gas. The meteorological measurements included air temperature, atmospheric pressure, wind speed and direction, and relative humidity. The radiation

measurements were made using a Kipp & Zonen NR-lite2 net radiometer positioned at 3 m above the ground with field of view at the grass field. The radiometer measures incoming and outgoing shortwave radiation that can be converted to photosynthetically active radiation (PAR), direct and reflected global radiation and net radiation. The soil surface heat flux was measured with Hukseflux HFP01 heat flux plate at a 5 cm depth. Annual precipitation was also reported from the nearby South African Weather Service (SAWS) Potchefstroom weather station (NCEI, 2015).

There were two separate measurements of soil moisture that were used to calculate stored soil water. The measurements of individual soil moisture sensors at depths 5, 20 and 50 cm (Delta-T ML2) were converted to average soil moisture using the weights of 150, 150 and 200 mm, respectively. These soil moisture measurements cover the complete experiment period from September 2010 to August 2016. Starting from March 2012 onwards, Delta-T PR2/6 probes were installed recording soil moisture at 10, 20, 30, 40, 60 and 100 cm depths. This profile measurement was converted to stored water using the weights

150, 100, 100, 150, 300 and 200 mm. The meteorological measurements were sampled every minute and 15 min averages were recorded. The meteorological variables were gapfilled using the marginal distribution sampling (Reichstein et al., 2005). The measurement site was visited once or twice a week during the measurement period to check the measurement status and correct errors if necessary. The measurement log was used during the data analysis phase to identify anomalies, outliers, or erroneous measurement periods. Further details about the site and the measurements are presented elsewhere (Aurela et al.,

2009; Räsänen et al., 2017).



## 2.3 Flux calculation and gapfilling

The details of the turbulent flux calculations are presented in Räsänen et al. (2017) but are summarized here for completeness. The turbulent fluxes were calculated as 30 min block averages after double rotation and the WPL density correction (Webb et al., 1980). The low frequency correction was performed according to Moore (1986) and high-frequency losses were corrected
using empirical transfer functions determined using sensible heat flux as a reference scalar. The sensible and latent heat flux values were discarded when the measured friction velocity $u_*$ was below 0.28 m s$^{-1}$, which was deemed as a state of low turbulence mixing. A steady state test was used to retain heat fluxes with less than 30 % and 100 % difference for gapfill model and final data, respectively (Aubinet et al., 2012). Latent heat fluxes were checked for acceptable range of $H_2O$ concentration and concentration variance to detect anomalous spikes due to condensation or rainfall. Heat flux values were filtered for outliers
by taking values for each month of all measurement years and filtering outliers using adjusted boxplot (Hubert and Vandervieren, 2008). The steady state check resulted in less than 30 % filtered fluxes, which were gapfilled using the marginal distribution sampling (MDS) from REddyProc package (Reichstein et al., 2005).

The inferred GPP was used to derive the water-use efficiencies to partition evapotranspiration into transpiration and evaporation. The measured NEE was partitioned into GPP and ecosystem respiration by using night-time mean respiration for
daytime respiration and calculating GPP as the difference between NEE and ecosystem respiration. The night-time mean respiration was used instead of exponential temperature function because only 2 % of the fitting windows had a linear or exponential relation between ecosystem respiration and soil temperature. The difference between monthly estimated transpiration with night-time mean and exponential temperature function is small (Fig. S1). The GPP fit parameters and the night-time mean respiration were calculated in a moving data window that was defined for each day with an initial length of 6
days. The moving window was expanded up to 20 days if necessary to include at least 50 measurement points. The measured NEE had one large 25 day gap in September 2013 and the fit parameters were linearly interpolated during this gap. The NEE was calculated using the same filters as the heat fluxes and more details of are provided elsewhere (Räsänen et al., 2017).

The potential ET (PET) was calculated using the Priestley-Taylor formulation given by (Priestley and Taylor, 1972)

$$PET = \alpha_{PT} \frac{\Delta}{\Delta + \gamma_p} (\text{Rn} - \text{G}), \tag{1}$$

where $\alpha_{PT} = 1.26$ is the Priestley-Taylor coefficient, $\Delta = de^*/dT_a$, $e^*$ is the saturation vapor pressure given by the Clausius-Clapeyron equation and evaluated at the measured air temperature $T_a$, Rn is net radiation, G is soil heat flux and $\gamma_p$ is the psychrometric constant. The energy balance closure (EBC) slope was estimate for each year by regressing all measured half-hour values of Rn-G against the sum of the measured latent and sensible heat fluxes for the same period.



## 2.4    Annual ET uncertainty

The u* threshold was estimated using a bootstrap technique from 200 artificial replicates of the dataset (Wutzler et al., 2018). The mean u* estimate value for the whole dataset was 0.28, and heat flux and NEE values were discarded when u* was lower than this limit. The 5th, 50th, and 95th percentile of the estimates are 0.27, 0.29 and 0.32, respectively. The dataset was u* filtered and gapfilled with these three u* limits. The annual u* uncertainty range was calculated for each year k as follows

$$E_{u*,k} = \frac{ET_{max,k} - ET_{min,k}}{ET_{median,k}}, \qquad (2)$$

where $E_{u*,k}$ is the u* uncertainty for year k in mm.

The MDS gap-filling algorithm estimates random error for each half hour value based on the variance of the observed LE with similar meteorological conditions in a moving window using

$$E_{rand,k} = \sum_{1}^{n} \sigma^2, \qquad (3)$$

where $n$ is the number of 30 min periods in year $k$. The total uncertainty of the annual ET was calculated by adding the random error and u* uncertainty in quadrature to yield

$$E_{tot,k} = \sqrt{E_{u*,k}^2 + E_{rand,k}^2}. \qquad (4)$$

## 2.5    Partitioning ET

One approach for partitioning ET to transpiration and evaporation is presented by Berkelhammer et al. (2016). Referred to as Berkelhammer method, it was applied to each year individually due to large changes in vegetation phenology between the years. The method assumes that when T=ET, ET is linearly related to $GPP \times VPD^{0.5}$ and T/ET ratio approaches 1 intermittently. To estimate T/ET value for each 30 min period, $GPP \times VPD^{0.5}$ is plotted against ET for each year. For each equal sized $GPP \times VPD^{0.5}$ bin, the minimum value is defined as 5th percentile of ET. A linear regression line of these bins defines the ET value for which T=ET. Any value falling below the line is considered to have T/ET equal to 1.0. For other points T/ET is defined as the ratio between the observed and the minimum ET (representing T):

$$\frac{T}{ET_{optimal}} = \frac{min_{GPP}||ET||}{ET_{flux}} \qquad (5)$$

where $min_{GPP}||ET||$ is the minimum ET value and $ET_{flux}$ is the observed ET value. The slope and y-intercept of the T=ET line define the inverse of water use efficiency for each year based on the half-hour data. The monthly T/ET values were





calculated by taking mean of all values within the month. The 30 min data points used for the T/ET estimation were filtered with additional quality criteria following Zhou et al. (2016). Only data points with measured ET and positive GPP and net radiation were used. Data from rainy days were excluded. Two other methods were also used to estimate T so as to select the method most applicable to water limited ecosystems. In the water use efficiency (uWUE) method, the T=ET line is fitted using

quantile regression with zero-intercept, referred to as (uWUEp) (Zhou et al., 2016). The uWUEp was defined by fitting all 6 years of 30 min data, resulting in uWUEp = 11.55 gC $hPa^{0.5}$ / kg $H_2O$. Then monthly ET to GPP × $VPD^{0.5}$ line is fitted using linear regression with zero-intercept (uWUEa) and the monthly T/ET value is the ratio of uWUEa and uWUEp. The difference between the Berkelhammer method and the uWUE method is in the fitting of the T=ET line and in the calculation of the monthly T=ET values. The ET was also partitioned using the Transpiration Estimation Algorithm (TEA) (Nelson et al., 2018).

The TEA is a random forest regressor that first isolates most likely periods of T=ET, and then trains on GPP and T relations during these periods.

## 2.6   Soil desorption

The mean daily estimate of evaporation calculated from the half-hour estimates of T/ET allow indirect testing of whether estimated cumulative E scales linearly with $t_d^{1/2}$, where $t_d$ is a single-event drydown duration in days. This scaling is expected

for what is termed as stage-2 evaporation rate starting from the day after the rainfall event. During this stage, the daily E is limited by soil moisture conditions and soil physical properties (desorptivity) described elsewhere (Brutsaert and Chen, 1995). The daily evaporation rate can be expressed as $E = (1/2)D_E t_d^{-1/2}$ and the cumulative daily E can be expressed as $D_E t_d^{1/2}$, where $D_E$ is to be determined. The expected range of $D_E$ based on several experiments is about 3 to 6 mm $d^{-1/2}$ as discussed elsewhere (Parlange et al., 1992). By regressing cumulative daily E inferred from the aforementioned partitioning methods

upon $\sqrt{t_d}$ for a single dry-down period, the soil desorptivity $D_E$ can be computed and compared to literature values. Dry season precipitation events were sampled from June to August each year with condition of at least 8-day long dry down and 0.02 $m^3 m^{-3}$ increase in surface soil moisture at 10 cm depth. The wet season precipitation events were searched from April of each hydrological year. In five hydrological years there was 8-days long dry down period after precipitation in April with continuous daily evaporation estimate. April is also the month with the lowest CV in monthly value of EVI excluding the dry season

months. By sampling from late wet season, any differences in soil surface conditions may be seen in the late wet season evaporation events if ambient atmospheric variables do not exert stronger controls on the soil evaporation (as expected in stage-2 evaporation). The first day of fitting of the soil desorption was set to a day when soil evaporation decreased. This varied from 2 days to 6 days after the rainfall event. In April, the soil is dry enough for stage-2 conditions unlike in mid wet season when P frequency is higher and surface soil is wetter. The stage-2 soil evaporation after precipitation events was

modeled with two different estimates of soil desorption. First, the soil evaporation was calculated using the aforementioned $D_e$ from regression of cumulative soil evaporation. This represents the eddy covariance scale and the calculated daily evaporation should match with the partitioned soil evaporation estimate if the conditions for stage-2 evaporation are met. The



second estimate is a linearized solution for soil desorptivity based on initial surface soil moisture conditions (Black et al., 1969)

$$D_{e,\theta} = 2(\theta_i - \theta_0)\left(\frac{\overline{D}}{\pi}\right)^{\frac{1}{2}}$$

(6)

where $\theta_i$ is initial soil moisture, $\theta_0$ is the soil moisture at surface and $\overline{D}$ is the weighted-mean diffusivity that was set to 394 mm$^2$ d$^{-1}$ (Brutsaert, 2014). The first day soil moisture value at 10 cm depth was used for $\theta_i$ and $\theta_0$ was set to zero as a first approximation.

### 2.7 Satellite data

Changes in vegetation state were quantified using monthly average of MODIS 16-day EVI with 250 m spatial resolution
(MOD13Q1, collection 6) (Didan, 2015). To compare estimated T/ET to variations in vegetation phenology, monthly average of MODIS 8-day LAI (MOD15A2H, collection 6) with 500 m spatial resolution was used to relate monthly T/ET to LAI. The EVI signal is a ratio of spectral bands whereas the LAI has correct units of foliage area per ground area.

### 2.8 Rainy season timing and green-up dates

The rainy season length was estimated based on climatological threshold of 5 % of the mean annual rainfall (Guan et al., 2014).
The start of the rainy season was defined as the day when cumulative rainfall of the hydrological year (September to August) reached the threshold value 27 mm based on the long-term mean annual rainfall (540 mm). Similarly, the end of rainy season was estimated as the first day when cumulative rainfall, starting backwards from end of hydrological year (August), reached the same threshold value. The tree green-up dates were estimated from the raw 16-day EVI time series (Archibald and Scholes, 2007). The tree green-up date was defined as the day when the EVI signal was higher than a moving average of previous four
time steps at the beginning of hydrological year. This is the time when the EVI time series experiences a sudden increase. To characterize early wet season (September to November) precipitation, the mean daily rainfall statistics are estimated using daily mean precipitation amount ($\alpha$) and daily mean storm frequency ($\lambda$). The daily mean precipitation amount is the mean precipitation of rainy days. The mean storm frequency was calculated as the inverse of the mean time between rainy days.

## 3 Results

### 3.1 Hydrological years

Hydrological years are defined from September to August and are referred to by the year in which they begin. In 2011, the annual precipitation was substantially (187 mm) lower than ET and the total rainfall time was 32 h shorter than in other years. The histogram of daily soil moisture increases shows that there were 5 extreme wetting events (soil moisture increase > 20



mm day$^{-1}$) in 2011, whereas other years had only one extreme event (Fig. S2). The measured site precipitation was higher than values at the nearby weather station except in 2011 (Table 1 and Fig. S3). This comparison suggests that the underestimation in 2011 may be due to poor performance of the tipping bucket sensor during heavy rain events. The years 2010, 2012, 2013 and 2014 were years with close to or above the long-term mean annual rainfall of 540 mm, while 2015 was

5 an extreme drought year in South Africa. In general, annual P was lower than annual ET for each year (Table 1). The annual change of soil water storage, based on 10 cm to 100 cm depth soil moisture measurements, varied from 1 mm to 14 mm. This variation is small compared to the variation in other water balance components. The annual energy balance closure varied from 0.75 to 0.85, and is comparable to those reported among FluxNet sites (Stoy et al., 2013; Wilson et al., 2002). The year 2011 had a 2-week dry spell at the end January, and 2015 had a dry spell at the end of November, impacting the local EVI minimum

(Fig. 1). In 2015, the grass experienced a dieback and regrowth, leading to the second peak in EVI. Although annual P in 2015 was 77 mm lower than the long-term mean, the rainy season was nearly twice as long as in other years (Table 2).

**Table 1. Annual sum of water balance components for each hydrological year (September to August). The transpiration and evaporation are calculated from monthly T/ET estimates. The precipitation value in parenthesis indicates the annual sum from**
15 **nearby weather station. The EBC-slope stands for the slope of the energy balance closure with ordinate defined by measured Rn-G and abscissa defined by the sum of the measured latent and sensible heat fluxes.**

| Year | P | ET | P–ET | T | E | $\Delta\theta_{1m}$ | T/ET | PET | $EVI_{max}$ | $P_{duration}$ | EBC-slope |
|------|---|-----|------|---|---|-----|------|-----|------|------|-----------|
|  | (mm) | (mm) | (mm) | (mm) | (mm) | (mm) |  | (mm) |  | (hours) |  |
| 2010–2011 | 574 (332) | 658 ± 16 | −84 | 362 | 295 | - | 0.55 | 1133 | 0.32 | 266 | 0.75 |
| 2011–2012 | 421 (560) | 608 ± 16 | −187 | 291 | 317 | - | 0.48 | 1123 | 0.30 | 121 | 0.80 |
| 2012–2013 | 614 (537) | 667 ± 14 | −53 | 352 | 314 | 14 | 0.53 | 1109 | 0.29 | 159 | 0.85 |
| 2013–2014 | 530 (509) | 600 ± 14 | −70 | 348 | 252 | 12 | 0.58 | 1039 | 0.31 | 153 | 0.81 |
| 2014–2015 | 582 (407) | 642 ± 17 | −60 | 344 | 297 | 3 | 0.54 | 1057 | 0.27 | 158 | 0.83 |
| 2015–2016 | 465 (437) | 463 ± 11 | 2 | 179 | 283 | −1 | 0.39 | 1038 | 0.22 | 157 | 0.84 |
| Mean | 531 | 606 | −75 | 313 | 293 | 5 | 0.51 | 1083 | 0.28 | 169 | 0.81 |
| SD | 68 | 69 | 57 | 64 | 22 6 |  | 0.06 | 40 | 0.03 | 45 | 0.03 |



**Table 2. Rainy season timing and tree green-up dates. The start day refers to day of hydrological year and the end day to the day of subsequent year. Early wet season period spans from September to November.**

| Year | Tree green-up | Start of rain | End of rain | Rainy season length | Percentage of rainy season T=ET values | Mean $\theta_{5cm}$ of T=ET | Early wet season $\alpha$ | Early wet season $\lambda$ |
|---|---|---|---|---|---|---|---|---|
| | (DOY) | (DOY) | (DOY) | (days) | | ($m^3 m^{-3}$) | (mm d$^{-1}$) | (storms d$^{-1}$) |
| 2010–2011 | 238 | 309 | 127 | 183 | 75 | 0.10 | 5.0 | 0.44 |
| 2011–2012 | 230 | 302 | 112 | 176 | 67 | 0.07 | 6.6 | 0.21 |
| 2012–2013 | 242 | 250 | 109 | 224 | 84 | 0.05 | 6.1 | 0.35 |
| 2013–2014 | 254 | 294 | 90 | 161 | 81 | 0.06 | 6.8 | 0.47 |
| 2014–2015 | 241 | 298 | 108 | 175 | 81 | 0.08 | 6.4 | 0.38 |
| 2015–2016 | 241 | 247 | 207 | 326 | 84 | 0.07 | 5.9 | 0.14 |



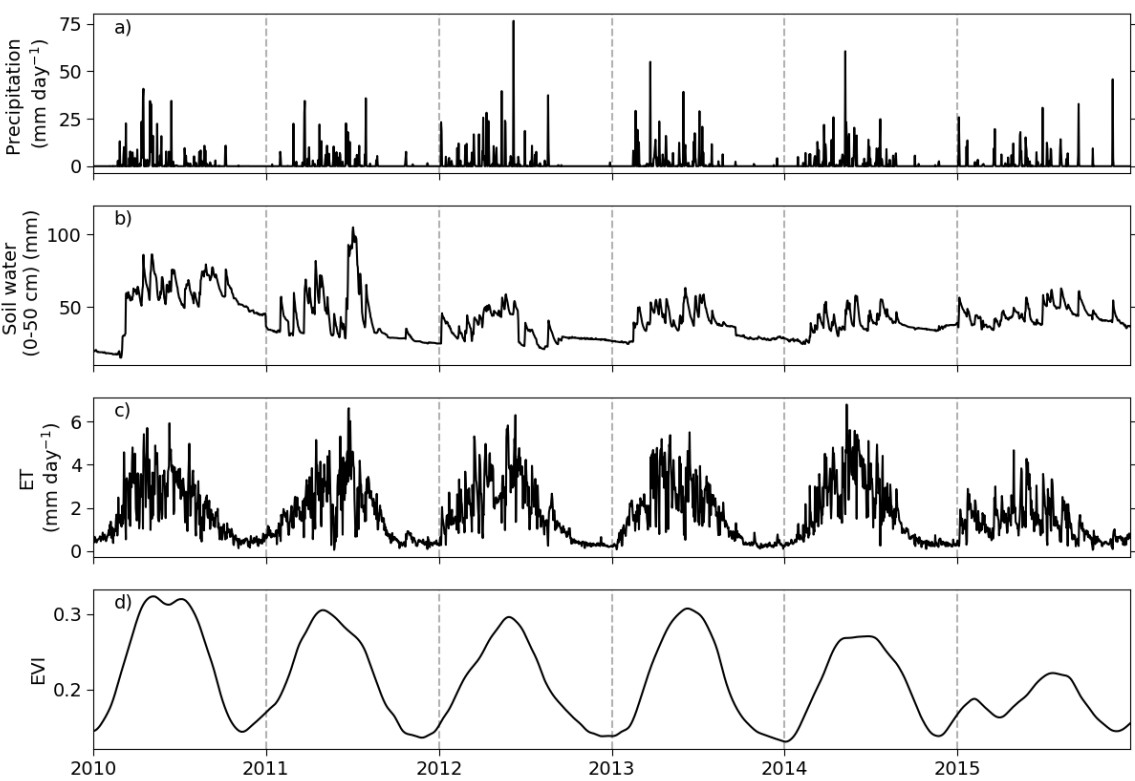

**Figure 1. Time series of daily precipitation, soil water (0-50cm depth), evapotranspiration and EVI.**

The estimated annual T/ET ratio varied from 0.39 to 0.58 (Table 1). The T/ET ratio is 0.09 less for 2015 than for 2011. For 2010, 2012, 2013 and 2014 the mean annual transpiration was $352 \pm 8$ mm year$^{-1}$. The shape of the cumulative monthly transpiration is similar for these years with 2014 having a late start (Fig. 2). In 2011, the cumulative transpiration is reduced compared to other years in December and January. In 2015, the cumulative transpiration increases linearly until April.





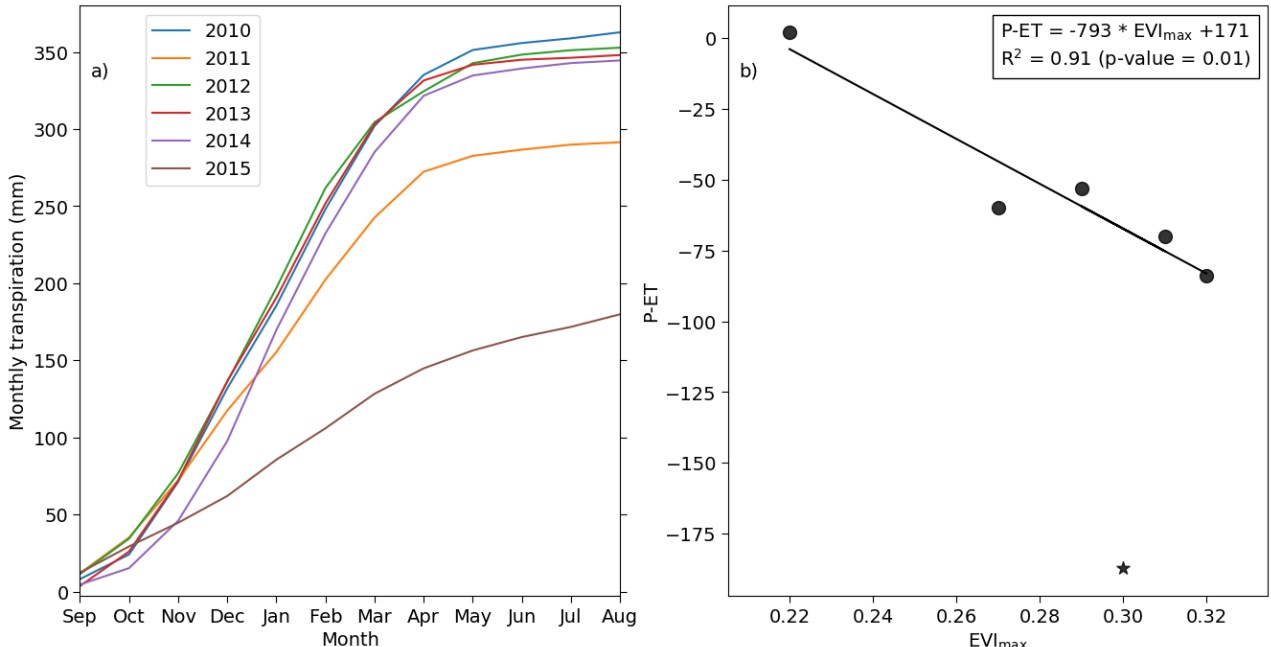

**Figure 2. a) Cumulative monthly transpiration for each hydrological year (September to August). b) Relation between annual maximum EVI and P–ET. The year 2011 (star) is not considered in the fit due to uncertain precipitation.**

The annual P–ET is not explained by the surface soil moisture storage changes ($R^2 = 0.51$, p-value = 0.28), but is inversely

related to the annual maximum EVI (Fig. 2). Using this relation, the estimated annual precipitation in 2011 would be 542 mm.

For the five normal years, the mean P–ET would be −67 mm. Also, the tree green-up days and start of the rainy period were

not linearly related (Table 2, $R^2 = 0.03$, p-value = 0.75), suggesting that the EVI signal rise at the onset of wet season is related

to changes in tree leaf phenology and not to grass LAI.





## 3.2 ET partitioning and monthly transpiration

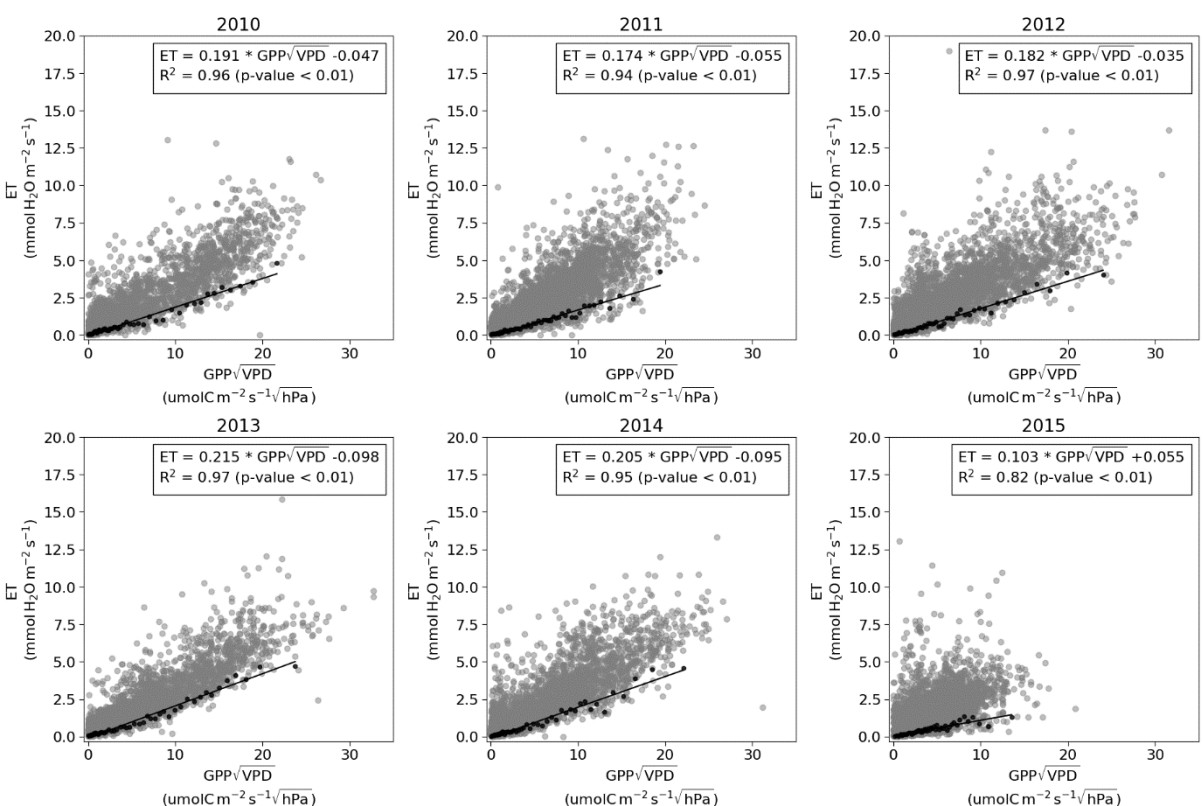

**Figure 3. Annual fitting of the T=ET line for each year. The x-axis is binned and one black dot is the minimum (5th percentile) of the bin. Linear regression is fitted to these minimum values. The grey dots indicate the half-hour data points.**

The bin values of the variable $GPP \times VPD^{0.5}$ were linearly related to 5th percentile of measured ET (Fig. 3) with the largest scatter during the drought year ($R^2 = 0.82$ in 2015). The $GPP \times VPD^{0.5}$ values vs. ET points are not similarly distributed every year. The years 2011 and 2013 had the same annual ET but there is more variation in ET values in 2011 for each $GPP \times VPD^{0.5}$ bin. The slopes of annual T=ET lines have more variation than annual transpiration. In 2010 to 2014, the annual T amount was

10 not significantly correlated with T=ET slope ($R^2 = 0.28$, p-value = 0.35). For all years, the mean surface soil moisture during T=ET instances is 0.1 $m^3 m^{-3}$ or less (Table 2). The slope and y-intercept of T=ET line are linearly related (Fig. 4). In addition, the slope of T=ET line is inversely related to the rainy season length when the year 2011 is excluded from the fit (Fig. 4). The slope represents the T=ET values, and in 2011 only 67 % of those values were from the rainy season, as opposed to ranging 75 % - 84 % in the other years (Table 2). A higher slope in 2011 value would mean lower water use efficiency. Nevertheless,

based on these relations, it is possible to estimate the annual T=ET line based on the rainy season length alone.





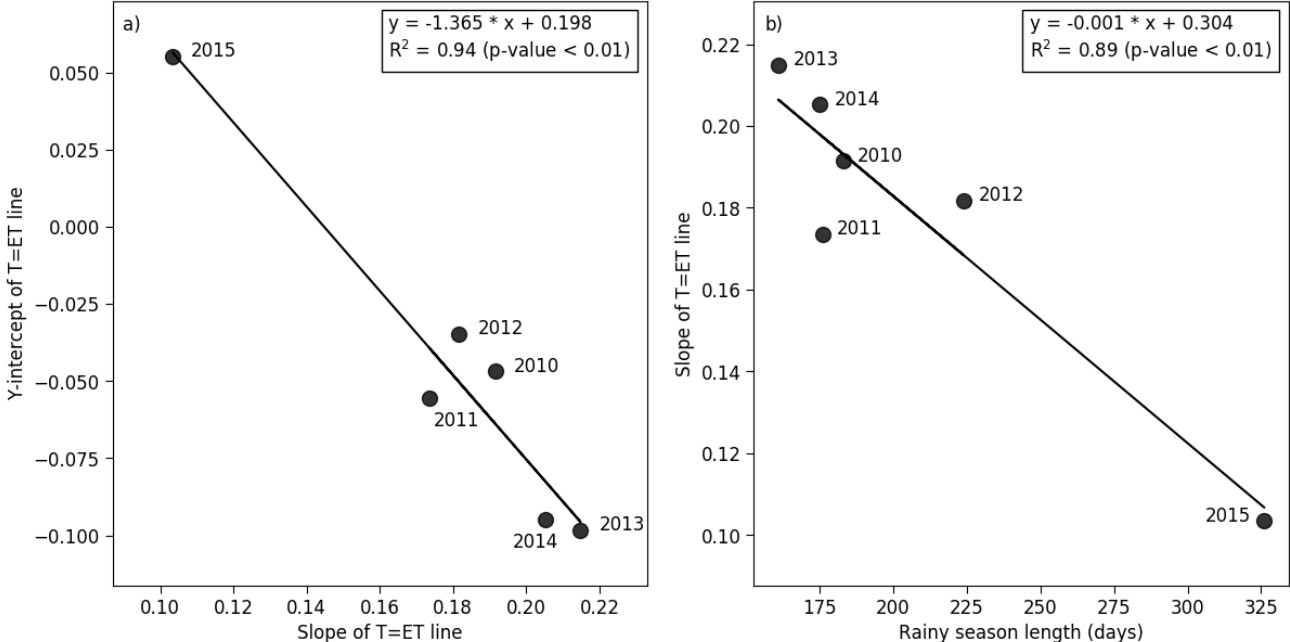

**Figure 4. a) Relation between slope and y-intercept of T=ET line. b) Relation between rainy season length and slope of T=ET line.**



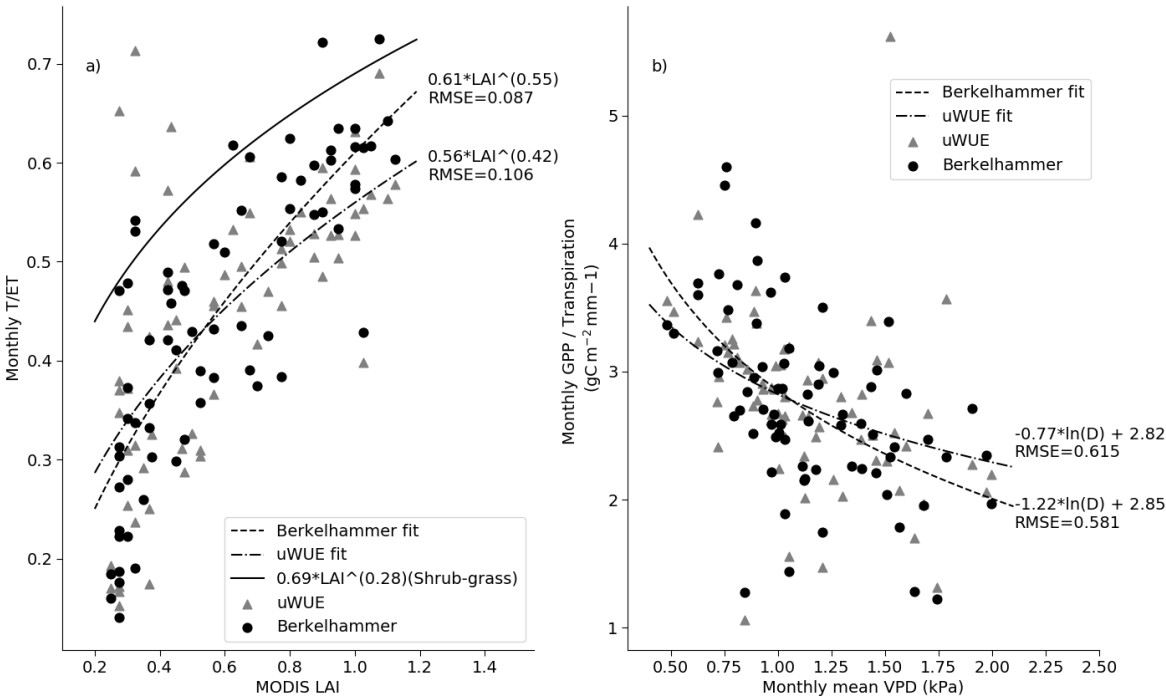

**Figure 5. (a) Relation between monthly MODIS LAI and T/ET. The black line is the relation between LAI and T/ET for shrub and grass ecosystems (Wei et al., 2017). (b) Relation between monthly mean VPD and monthly GPP/Transpiration. The dashed line is the fit for T/ET estimated by Berkelhammer method (black dots) and the dash-dotted line is the fit for T/ET estimated by uWUE method (grey triangles).**



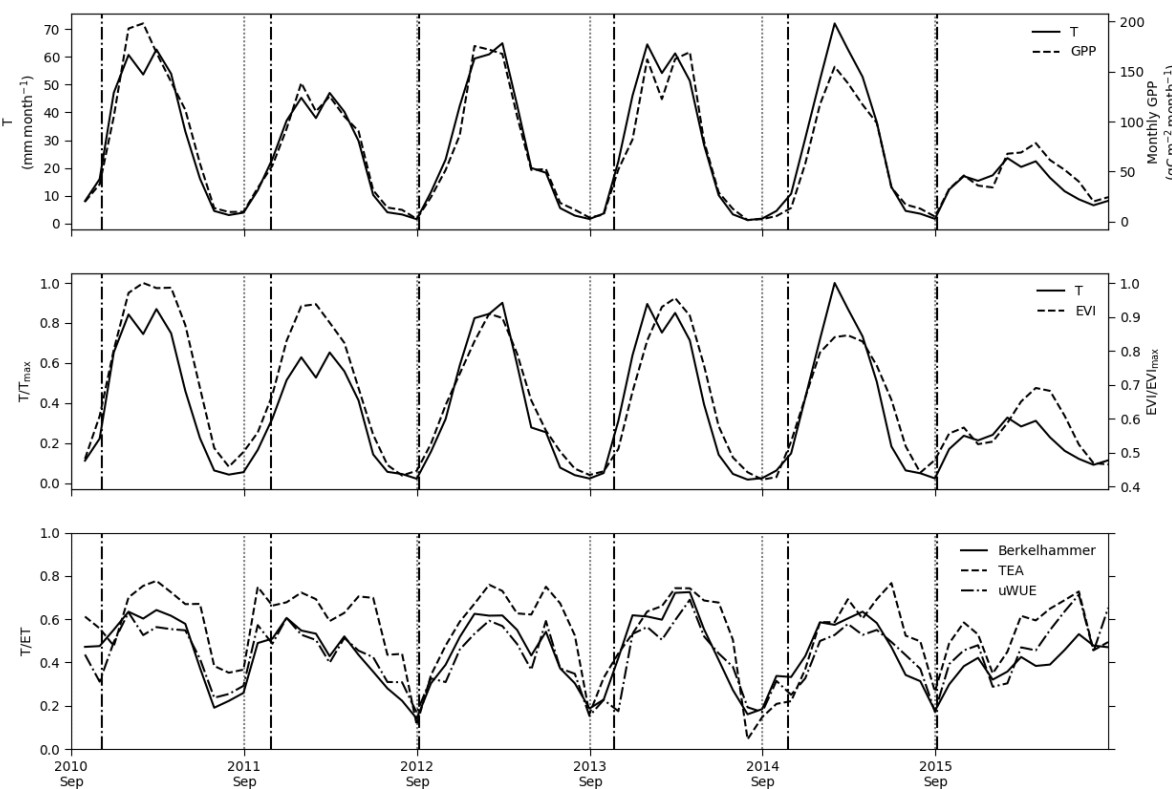

**Figure 6. Time series of monthly transpiration, EVI and T/ET. The T/ET ratio was estimated with Berkelhammer, uWUE and TEA methods. The dotted vertical line indicates start of the hydrological year (September 1st) and the dash dotted line indicates start of rainy season.**





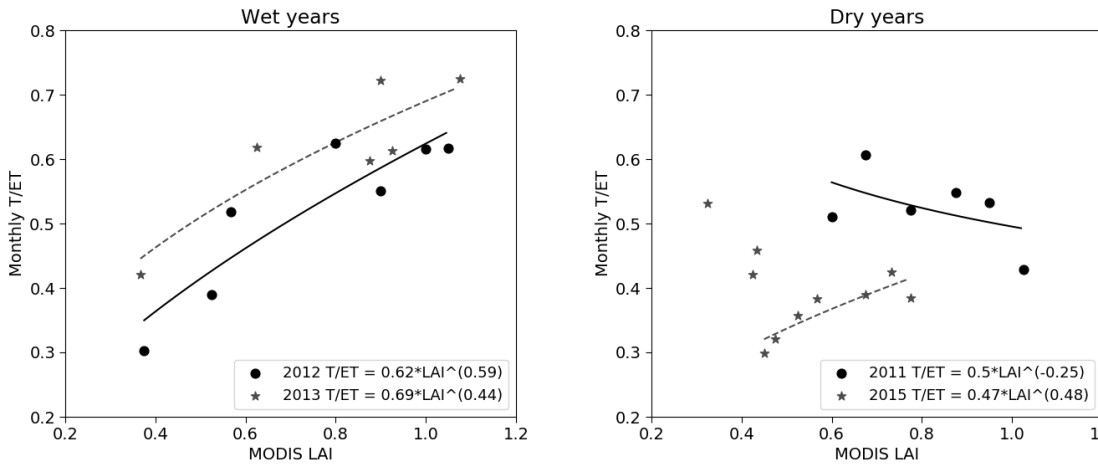

**Figure 7. Rainy season relation between monthly MODIS LAI and T/ET (Berkelhammer method) for wet (2012 and 2013) and dry (2011 and 2015) years. The year 2013 had higher early wet season precipitation than year 2012. In 2015 the November, May and June values were excluded as outliers from the fit.**

The comparison of the three different monthly T/ET estimates shows that T/ET based on the TEA method is consistently higher than the other methods during the wet season (Fig. 6). The annual maximum T/ET by this method has small variance compared to that obtained by the other methods. The root-mean squared error of the LAI to T/ET relation is lower for the Berkelhammer method than for the uWUE method (Fig. 5). The uWUE method has high T/ET values at low LAI, which

corresponds to the 2015 dry season values. These values exceed the values of a multi-site derived shrub-grass LAI vs. T/ET relation (Wei et al., 2017). The VPD response of monthly GPP/T of the Berkelhammer method is more non-linear than the uWUE estimate (Fig. 5). At the end of 2013 and early 2014, the monthly GPP/T was low (around 1.3 gC m$^{-2}$ mm$^{-1}$) and insensitive to changes in VPD.

The largest difference between uWUE and Berkelhammer T/ET estimate are during 2015 from March to June. During that

period, monthly GPP decreased and T/ET increased based on all methods, but the increase based on the Berkelhammer method is small relative to the others. In June 2016, Berkelhammer-based T/ET was 25 % lower than uWUE T/ET for EVI value that is half the typical wet season maximum value. The T/ET values in the late wet season of 2015 based on both TEA and uWUE methods are likely an overestimate, given the decrease in GPP and low EVI values during that drought year.

The annual T/ET is higher for years with more frequent early wet season precipitation (Table 1 and 2). During the rainy season

the T/ET is consistently higher in 2013 than in 2012 for the same LAI range, and explained by the higher precipitation frequency in 2013 (Fig. 7). During the drought year, dry season T/ET values are higher than wet season values due to significant precipitation during these months. In 2011 the tree green-up date was the earliest and the T/ET value at the start of the rainy season was the highest and close to annual maximum, and thus the rainy season relation between LAI and T/ET was a decreasing curve.





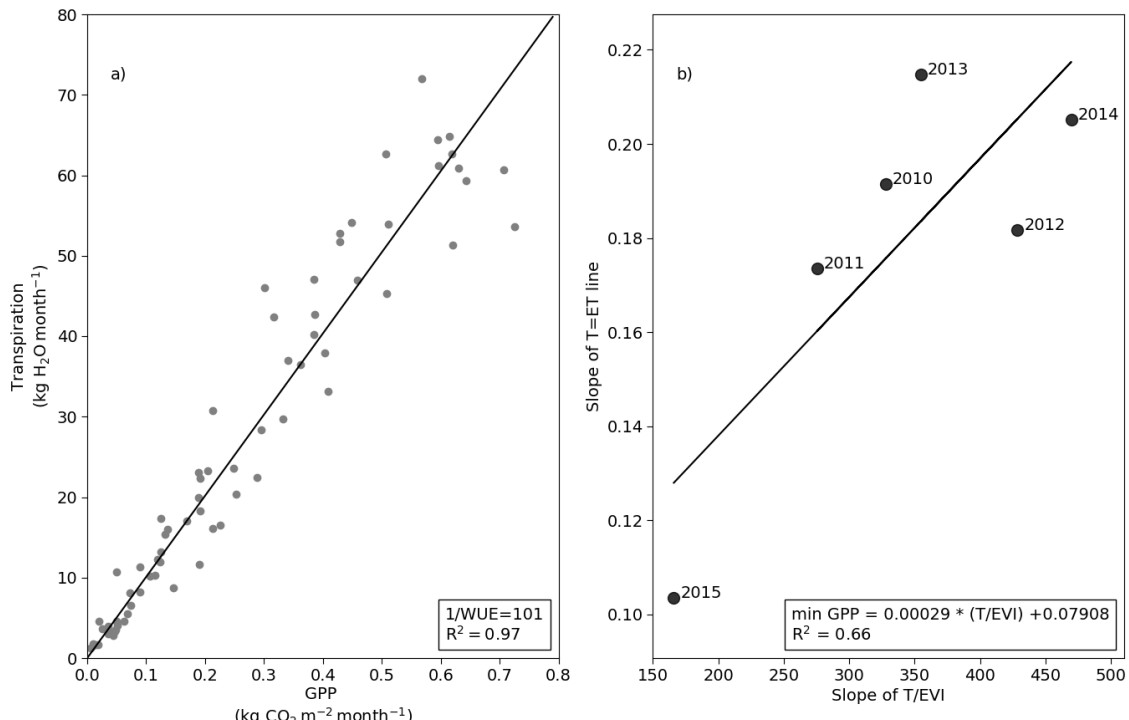

**Figure 8. (a) Inverse of water use efficiency estimated from monthly transpiration and GPP with zero intercept. (b) Relation between annual slope of T/EVI (Fig. S4) and slope of T=ET line (Fig. 3).**

The monthly T and GPP were linearly related (Fig. 6, $R^2 = 0.93$, p-value<0.01) and it enables an estimate of an effective

(constant) ecosystem water use efficiency using a zero intercept regression. The inverse of the constant water use efficiency

was 101 g $H_2O$/g $CO_2$ (Fig. 8). The mean annual fitted inverse of the water use efficiency was $100 \pm 13$ g $H_2O$/g $CO_2$.

Similarly, monthly EVI and T were linearly related ($R^2 = 0.86$, p-value < 0.01). In 2013, the monthly EVI was lagging the

estimated monthly T (Fig. 6). This is also the year when early wet season daily mean precipitation amount and storm frequency

are the highest (Table 2). In 2010, the early wet season storm frequency is high and the annual EVI is the highest but the EVI

does not lag T. The annual slope of T/EVI was linearly related to the slope of the T=ET line (Fig. 8). This means that for a dry

year, the transpiration is smaller per unit of EVI than for a wet year, resulting in a lower slope of the T=ET line.

### 3.3 Dry season water balance and soil desorption

Due to the abnormal rainfall timing in 2015, the dry season precipitation was higher than the dry season ET, opposite of the

other years (Table 3). The average dry season transpiration was 11 mm $(3 \text{ months})^{-1}$ (Table 3). This estimate suggests a

minimum tree transpiration of 44 mm $\text{year}^{-1}$. The estimated cumulative E scales linearly with $t_d^{1/2}$ ($R^2 > 0.97$) during five





late wet season events and four dry season precipitation events allowing us to check estimated E dry down trend to expected stage-2 evaporation and compare to soil moisture based estimate of E (Fig. 9, Table 4). For the dry season 2012 there was no precipitation event during June to August, and in dry season 2013 there was one precipitation event but the soil moisture at 10 cm did not register that event. For these years, the dry season soil desorptivity was not estimated. In 2010 late wet season there

was no 8-day dry down period and hence no estimate of desorption was possible for late wet season 2010. The late wet season soil desorption estimated from partitioned evaporation ($D_e$) ranged from 2.6 to 7.1 mm d$^{-1/2}$. The mean soil moisture based $D_{e,\theta}$ ranged from 2.0 to 3.5 mm d$^{-1/2}$. The late wet season soil desorption ($D_e$) increases linearly with increasing initial daily evaporation (Table 4, $R^2 = 0.86$, p-value<0.05) and for all events ($R^2 = 0.54$, p-value<0.05). The $D_e$ values were not correlated with first day air temperature or water vapor deficit but there was significant correlation with soil moisture at 20 cm

depth ($R^2 = 0.63$, p-value<0.02). The wet season $D_{e,\theta}$ estimated from soil moisture is similar with $D_e$, except in 2010 and in 2013 when precipitation event was only 6.5 mm. For 2014 and 2015 the dry season $D_{e,\theta}$ is higher than $D_e$. One possible explanation for this difference is a potential drift in the dry season surface soil moisture sensor (due to changing sensor-soil contact). Evidence supporting this modest drift is that the minimum soil moisture value is approximately 0.04 higher in later years than in 2011. The much larger $D_{e,\theta}$ in 2015 shows that the 'small scale' estimate can be much larger than the eddy

covariance scale estimate although the overall mean $D_e$ is higher than $D_{e,\theta}$. Despite different initial conditions primarily controlled by the first day E, the wet and dry season estimated soil desorption are in similar range, and the $D_e$ estimate of wet season dry downs matches estimated daily E from the partitioning methods (Fig. 9). The lowest wet and dry season $D_e$ values where estimated during the drought year 2015, characterized by grass regrowth and reduction in annual transpiration.

**Table 3. Dry season (Jun-Aug) sum of water balance components.**

| Dry season | P | ET | T | E |
|---|---|---|---|---|
| | (mm) | (mm) | (mm) | (mm) |
| 2010 | 22 | 52 | 11 | 41 |
| 2011 | 17 | 40 | 8 | 31 |
| 2012 | 2 | 33 | 10 | 22 |
| 2013 | 6 | 29 | 6 | 23 |
| 2014 | 12 | 34 | 9 | 24 |
| 2015 | 60 | 47 | 23 | 24 |



**Table 4. Late wet season (April) and dry season soil desorption estimated from partitioned evaporation ($D_e$) and from surface soil moisture ($D_{e,\theta}$). E_day1 is the evaporation of the first day of soil desorption fit.**

| Year | Start date | P amount | $D_e$ | $D_{e,\theta}$ | $E_{day1}$ | $\theta_i$ |
|---|---|---|---|---|---|---|
| | | (mm) | (mm d$^{-1/2}$) | (mm d$^{-1/2}$) | (mm) | (m$^3$m$^{-3}$) |
| Wet season | | | | | | |
| 2011 | 2012-04-01 | 45.0 | 7.1 | 3.1 | 2.7 | 0.14 |
| 2012 | 2013-04-26 | 46.2 | 3.7 | 3.5 | 1.9 | 0.16 |
| 2013 | 2014-04-24 | 6.5 | 3.0 | 2.0 | 1.8 | 0.09 |
| 2014 | 2015-04-26 | 13.6 | 3.5 | 2.8 | 1.5 | 0.13 |
| 2015 | 2016-04-23 | 10.8 | 2.6 | 2.7 | 1.1 | 0.12 |
| Dry season | | | | | | |
| 2010 | 2011-06-09 | 21.4 | 2.9 | - | 0.5 | - |
| 2011 | 2012-06-24 | 13.6 | 2.7 | 2.1 | 0.5 | 0.10 |
| 2014 | 2015-09-06 | 41.2 | 4.3 | 4.6 | 0.8 | 0.21 |
| 2015 | 2016-07-26 | 47.4 | 2.3 | 5.2 | 0.8 | 0.25 |

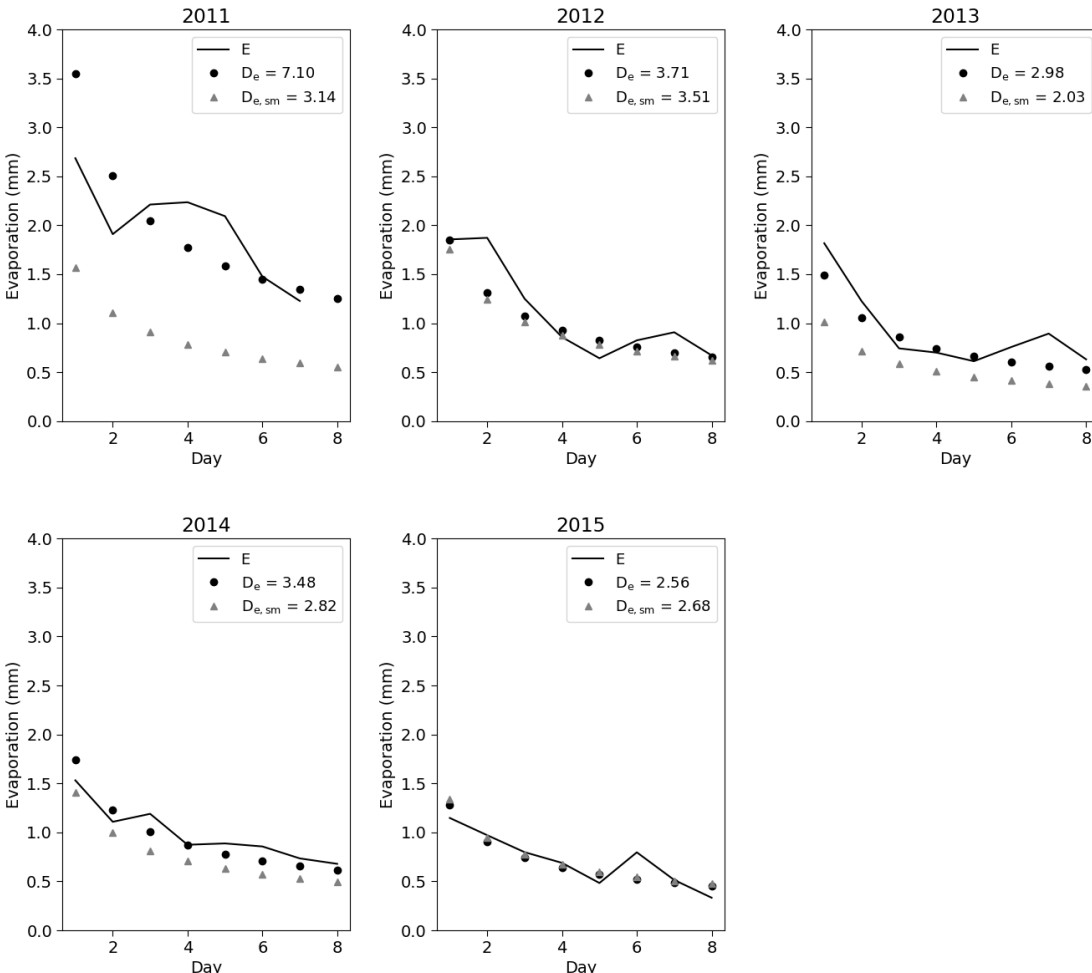

**Figure 9. Daily evaporation after rainfall event in April except in year 2010. The line is estimated daily evaporation from partitioning of ET. The dots indicate daily evaporation estimated from cumulative daily evaporation and triangles indicate daily evaporation estimated from initial surface soil moisture.**



## 4    Discussion

The annual T was highly constrained to $352 \pm 8$ mm year$^{-1}$ for four wet years with rainfall around or above long-term mean ($P \geq 540$ mm). This is consistent with small interannual variation in water use of the deep-rooted trees, and moderate water

stress of $C_4$ grass, potentially aided by grazing pressure that limits the grass leaf area. During a drought year, the transpiration was reduced due to grass dieback-regrowth and possibly due to other changes in soil surface properties (cracks in surface crust) that enhanced evaporation.

Roots of the dominant tree species, *Vachellia erioloba*, are deep, and this site is at the wet end of its distribution. The estimated radiocarbon age of these trees is approximately 20 years (Steenkamp et al., 2008). The roots of *Vachellia erioloba* have been

reported to extend up to 60 m depth (Jennings, C.M.H., 1974), and the species was shown to take 37 % of its water below 1 m depth (Beyer et al., 2018). In addition, the horizontal extent of this species' roots can exceed 20 m (Wang et al., 2007). While installing the soil moisture profile measurements at the site, tree roots were observed at 40 cm depth and 15 m away from the nearest tree. This suggest that trees at the site are likely decoupled from recent precipitation and shallow soil moisture, and that information from other studies can be used to compare to annual tree transpiration at the site. For example, in a similar

site at this region (P=241 mm) the ratio of annual/(dry season) transpiration of *Vachellia erioloba* (5 m height) was 3.9/0.6 mm d$^{-1}$ (Tfwala et al., 2019). Multiplying this ratio with our 11 mm dry season tree transpiration results to 72 mm of annual tree transpiration. This is similar to the 87 mm yr$^{-1}$ annual transpiration based on sap flow measurements conducted in a site of deep rooted *Vachellia tortillis* trees in Senegal, comprising 11 % of the ground cover (P=280 mm; Do et al., 2008). We note that a study of global pine plantation showed that, sap-flux-based annual transpiration of trees with access to sufficient soil

moisture can be estimated as 0.55 mm d$^{-1}$ * LAI (Tor-ngern et al., 2017). Based on woody species LAI in four seasons (1.12, 0.12, 0.3 and 0.53) of one year at the site (Räsänen et al., 2017), annual pine transpiration at the site would be 102 mm, not much higher than estimated for the native trees in Senegal. The interannual variation in annual T based on sap flow measurements has been small. Finally, at a site of shallower rooted trees, at Nylsvley, South Africa, savannah of higher ET/P ratio, trees comprising 30 % cover transpired 126 mm annually based on measurements and modelling. Adjusting this estimate

by fractional cover would yield 63 mm for our 15 % tree cover. Based on these estimates, the annual tree transpiration at the site may range from 60 to 90 mm.

A hypothetical explanation for the small interannual variation in total transpiration during wet years is that the $C_4$ grass was experiencing only moderate water stress. The reduction in photosynthesis in $C_4$ grass is more related to non-stomatal limitations compared to $C_3$ grass, which is predominantly limited by stomatal control (Ripley et al., 2010). In South African field

conditions over one growing season, the $C_4$ grass had tendency to maintain constant difference between predawn and midday leaf water potential, and transpiration was similar at rain-fed and irrigated pots (Taylor et al., 2014). The seasonal course of transpiration at our site was similar in wet years and only shifted in time due to different start date of the rainy season. A


rainfall timing experiment with $C_4$ grass in growth chambers showed that there was no difference in the biomass for grass grown in frequent light shower and in intermittent storm every 12-day rainfall (Williams et al., 1998). It is likely that the variability in the rainfall timing in wet years did not produce water stress in the $C_4$ grass layer, resulting in small interannual variation of $C_4$ grass transpiration.

The invariability of ecosystem scale transpiration was observed in many forest ecosystems with a mixture of species of different drought response (Oishi et al., 2010). The mechanism for the invariance in these forests is related to observed higher VPD during drought years. In contrast, at Welgegund the water availability for grass is the determining factor of transpiration. Transpiration is constant for wet years with frequent early season rainfall but during drought the grass undergoes a dieback-regrowth period during mid wet season, resulting in reduced transpiration. During this period, bare soil cover, soil surface

properties and increased proportion of tree transpiration of the total transpiration affect the monthly transpiration. It is difficult to generalize if this invariance of annual transpiration would hold for sites with higher grass LAI, and where grass and tree roots share the same soil volume.

At monthly scale, transpiration and GPP were linearly related, producing a near-constant inverse water-use efficiency estimated at $100 \pm 13$ g $H_2O$/g $CO_2$. This is relatively close to the field scale long-term grass community value of 127 g $H_2O$/g

$CO_2$, but different from the combined tree and grass value of 420 g $H_2O$/g $CO_2$ for the shallower rooted trees and 30 % tree cover savanna at Nylsvley, South Africa (Scholes and Walker, 1993).

### 4.1   Uncertainty

Wind-induced underestimation is a well-known problem in rainfall measurements. A comprehensive study estimated the underestimation for our rain gauge (Casella) at 0.5 m height to be 9.4 % at the measurement site with similar mean wind speed

(5 m s$^{-1}$) and annual rainfall (P=700–1000 mm) (Pollock et al., 2018). At higher wind site the underestimation was 11 % at 0.5 m height and 17 % at 1.5 m height. The 9.4 % underestimation can be considered minimum and annual P adjustment (i.e. 1.094 measured P) would lead to annual mean P$\approx$ ET for five years excluding the year 2011. A 15 % underestimation would result in a positive P–ET for the five years.

The ratio of annual ET uncertainty to annual ET was 2 %, lower than 5 % to 9 % range reported from eddy covariance ET

measurements from cultivated area in Benin (Mamadou et al., 2016). The difference may be due to different error terms in the uncertainty estimation. At this site, the mid dry season ET ranged from 45 to 67.5 mm (3 months)$^{-1}$ (mid dry monthly value multiplied by three) in the cultivated site that has isolated trees (height < 10 m) and bare soil during the dry season (Mamadou et al., 2014, 2016). This is higher than the range of 29 to 52 mm measured here. A possible explanation is the relatively high water table at Benin site (depth is 3 m during the dry season at the Benin site) and its higher annual precipitation (P=1200

mm).





## 4.2    Remote sensing of T/ET

The difference in annual T/ET during the wet years was ≤ 0.09, which is an upper limit for a given ecosystem from a joint modeling and observation analysis (Paschalis et al., 2018). The comparison between the wet years with contrasting precipitation frequency suggest that higher precipitation frequency leads to higher monthly T/ET for the whole rainy season.

However, the seasonal course of T/ET was different in the two years with lower annual T/ET. The earliest tree green-up resulted to high T/ET at start of the rainy season and an early decline in T/ET. The anomalous timing of rainfall in the drought year resulted to higher T/ET during the dry season. For these reasons, during years with higher water stress accompanied by higher relative contribution of tree transpiration, the relation between monthly LAI and T/ET maybe altered from the expected nonlinear response.

For these water-stressed years, the partitioning of tree and grass contribution to LAI and T/ET may be needed in order to derive meaningful relationships at monthly scale. The analysis here does not reveal whether most of the deviations in the LAI-T/ET relation is due to grass or increasing role of bare soil evaporation and thus surface heterogeneity. A recent study used a canopy height model, Sentinel vegetation indexes (10 m spatial resolution) and Sentinel radar band, to analyse the 2015 drought in Kruger national park (Urban et al., 2018). These new remote sensing products combined with spatial measurements of surface

soil conditions and tree transpiration may enable to partition LAI into grass and tree components. The effect of dieback-regrowth to annual transpiration is also interesting because a stochastic model based on measured precipitation statistics with explicit bare soil, grass and tree cover showed that vegetation dynamics had little effect on annual transpiration and production (Williams and Albertson, 2005).

## 4.3    Soil desorption

Despite majority of the T=ET moments concentrate at the wet season, the analysis of soil desorption from wet and dry season shows that the estimated daily soil evaporation under stage-2 condition has the expected characteristics of diffusion limited soil evaporation. The experimental values of the initial stage-2 evaporation vary from 1 to 3 mm day$^{-1}$ for various soils and boundary conditions (Shokri et al., 2009). The daily evaporation rate was less than 2.7 mm d$^{-1}$ on the first day of all wet season drying events and below 1 mm d$^{-1}$ of all dry season events, which means that evaporation was stage-2 in all the drying

events considered here. The differences in $D_e$ values were not explained by ambient meteorological conditions but higher first day evaporation and initial soil moisture at 20 cm resulted to higher $D_e$. In laboratory conditions with full wetting of sandy soil columns, the stage-2 evaporation was shown to increase with ambient temperature (Ben Neriah et al., 2014). In the lab, the soil are dried homogeneously and continuously whereas in field conditions there can be large variance in surface soil conditions and soil characteristics. The first day evaporation control of $D_e$ is expected due the large spatial scale of the eddy

covariance measurement, whereas small scale ambient measurements may not explain average evaporation of large spatial extent. However, the significant correlation with soil moisture at 20 cm suggest that the soil moisture at this depth is a better representative of the column average soil moisture than the soil at 10 cm depth. The slightly lower values of soil desorption





during dry season compared to wet season are explained by the lower first day evaporation and thus give confidence that the estimated evaporation in dry season is also reasonable.

## 4.4    ET partitioning methods

The Berkelhammer and uWUE transpiration estimates were closer to reported grassland T/ET values and more similar than the TEA estimate. The largest difference between Berkelhammer and uWUE methods were in late wet season and the dry season of the drought year. During that time, the uWUE and TEA estimated T/ET values were high (up to 0.7) while GPP and EVI were low. For the uWUE method, one-to-one T=ET line is fitted using quantile regression for all six years' data combined, and the intercept is forced through zero. Aperiodic, single T=ET line rather than annual line, and the quantile regression are likely the reason for the difference between the uWUE and Berklhammer method. On the other hand, the TEA algorithm had an unusually constant annual maximum T/ET and higher wet season T/ET estimates than the other two methods. The algorithm does not use measured soil moisture in the training period, and instead uses simple P and ET water balance (Nelson et al., 2018). This is the most likely reason for small interannual variance of the maximum T/ET values of the TEA method.

The low surface soil moisture values during T=ET periods and their concentration in the rainy season give confidence that the annual fitted T=ET lines correspond to periods when T equals ET. The water balance analysis focused on monthly and annual timescales when the ET partition methods have shown good agreement with independent estimates (Berkelhammer et al., 2016; Zhou et al., 2018). Berkelhammer showed that a 3-day running mean of the half-hour T/ET estimates reduced the root-mean-square difference to $\leq 0.2$ between the Berkelhammer method and isotopic estimate of T/ET (Berkelhammer et al., 2016). Therefore, the random error of the monthly means of the half-hour T/ET estimates in this study is assumed to be small. For a Mediterranean tree-grass savanna, it was shown that T/ET was rarely over 0.8 (Perez-Priego et al., 2018). In contrast to the Mediterranean site, the Welgegund site has sandy soil, deep rooted trees and no clay horizon close to surface soil. More importantly, the mean surface soil moisture was 0.1 $m^3 m^{-3}$ or below for the half-hour moments when T=ET. This low soil moisture resulted in small diffusion limited soil evaporation and thus periods when T equals ET. This is another independent confirmation of the partitioning of ET to E and T (even at such short time scales).

## 5    Conclusion

The analyses shows that, on annual time scales, transpiration appears to be conservative at the grazed grassland savanna with deep rooted trees. The grazing limits maximum grass biomass and few mature trees are small but constant users of soil water from deeper layer. The tree transpiration is, to large extent, decoupled from short-term precipitation variability. The grazing pressure and moderate water stress of C$_4$ grass lead to nearly constant C$_4$ grass annual transpiration during the growing season. During drought caused by intermittent rainfall timing, the annual transpiration is reduced due to grass dieback-regrowth that alters temporal dynamics of bare soil cover and infiltration, and complicates monthly T/ET relation to LAI. The precipitation timing can alter annual growth of C$_4$ grasses, that tend to maintain constant transpiration, but react quickly to severe drought,

potentially leading to large reduction in annual transpiration. However, annual ET remains approximately equal to P even during drought due to increased soil evaporation. Thus, at annual scales, ET ≈ P, and annual T is roughly conservative despite large variations in annual P. The work here showed that the conservative nature of T is not simply due to the presence of trees accessing deep water. The dynamics of $C_4$ grasses and their water use efficiency also contributes to a conservative T. Although

further work is required to assess the generality of these conclusions to other savanna systems, the methodologies developed and tested here can be employed in investigating a wide range of arid and semi-arid ecosystems experiencing water limitations.

**Data availability**

The data used in this study are available online https://doi.org/10.6084/m9.figshare.11322464 (Räsänen et al., 2019).

**Author contributions**

MR, RO and GK designed the analysis; VV, MA, and JT did the EC data processing. MA, VV, PB, JT, PVZ, MJ, SS, TL, LL, MK, JR conducted the measurements.  All authors contributed to the final version of the manuscript.

**Competing interests**

The authors declare that they have no conflict of interest.

**Acknowledgements**

This work was supported by the Finnish Meteorological Institute, North-West University and the University of Helsinki, and Finnish Academy project Developing the atmospheric measurement capacity in Southern Africa and the Finnish Centre of Excellence, grant no. 272041. Financial support for R. Oren was provided by the Erkko Visiting Professor Programme of the Jane and Aatos Erkko 375th Anniversary Fund, through the University of Helsinki. G. Katul acknowledges partial support from the U.S. National Science Foundation (grant numbers NSF-AGS-1644382 and NSF-IOS-175489). The authors thank the

farmers at the ranch for their help in the setup and instrument maintenance.

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
