# Peer review of "The effect of rainfall amount and timing on annual transpiration in grazed savanna grassland"

_Hydrology and Earth System Sciences, 2019_

## Referee Comment (RC1) · Anonymous Referee #1 · 6 Mar 2020

Dear Matti Räsänen and co-authors,

Thank you for the article about the effect of rainfall amount and timing on annual transpiration in the grazed savanna grassland.

This is an extremely interesting paper on a very important subject in an overall data-scarce region. I believe that overall the methods and scientific work is sound, however I do think that the paper would benefit from some editing and streamlining. Often sentences are long and cumbersome and the presentation seems redundant and not distinct. Thus most of my feedback is on presentation quality and not scientific quality or significance. I believe it does present a substantial contribution to scientific progress within the scope of Hydrology and Earth System Sciences in terms new concepts, ideas, methods, and data. I believe the scientific approach and applied methods to be

valid. I believe that the results are discussed in an appropriate and balanced way.

They consider some related work but have some pretty large omissions. They only invoke one other, non-directly-related, study using eddy-covariance in Africa. This study was in Benin, in a very different climate zone. Meanwhile, if you include the whole African continent, there are have been other studies using eddy-covariance, many in more similar climatic areas. It would be interesting to compare your results with those of more similar studies, for example see the work of Marc Parlange's group in Burkina Faso, or group by the French CIRAD organization and associated researchers throughout francophone west Africa, or Kelly Caylor's work in East Africa, or some older studies in Niger and Nigeria.

One very minor, but very significant change in presentation would be to format paragraphs as distinct. I found it very hard to read because the appearance is of one single continual paragraph. Perhaps this is the flaw with the HESS template, but a compromise should be found!

Additionally, although the information that I would want from a picture of the site and a map is present (geographical coordinates, density of trees), I think it would be a tremendous boost to the ability of the reader to visualize if at least one figure was added including a map, a picture of the measurement station, a picture of the land cover (topography) around the station, and perhaps a sketch (to scale) of the different components of evapotranspiration (i.e. from C3 grasses, C4 grasses, woody-vegetation, ground water depth, soil layers, and any surface water etc. ).

I rate this paper Good (2) because I don't think the results and conclusions are currently presented in a clear, concise, and well-structured way. Some figures could be integrated with each other and of higher quality. The English is correct, though I think some editing (fewer words) would make it more readable.

The paper addresses (scientific questions):

Objective: to partition measured ET Quantify effect of seasonality on grass activity (this is very vague, and it is not clear why this is important).

How ET and its components vary according to mean annual P and vegetation change in a grazed savanna grassland ecosystem 3 study objectives: 1) quantifying the variation in P, ET, and estimated T based on met measurements at this new long term monitoring site 2) identifying the main drivers of the annual, seasonal, and monthly variation in the water balance components 3) identifying remote sensing variables that explain variations in T/ET and transpiration

Later the goal is to select the method most applicable to water limited ecosystems.

I think the paper would be strengthened if they adhered more to these goals and used them to structure the results, discussion, and even conclusions. I will try to reproduce the calculation with a similar data set, and I currently believe that you present ample information even if I have not yet tried. The title clearly placed annual transpiration as the goal. It does not invoke the three methods. The title is acceptable but you could make it more exciting or intriguing. Overall, I think they should write out abbreviations and equations more often in full text. Should any parts of the paper (text, formulae, figures, tables) be clarified, reduced, combined, or eliminated?

I found some sentences to be repeated more or less word for word in multiple parts of the text. This is completely unacceptable. Please respect your reader who reads all the way through and reword. Each sentence should bring new information. i.e. the first sentence of the introduction is the same as the short summary. For the short summary, I recommend saying some unknown problem and generalizing more. In general, sentences are jargon-y & convoluted. Some frames that could be said differently are repeated. I.e. partitioning? You could alternate it with divided, split, etc and talk about it more concretely. There are places where the article is missing. There are numerous places where a superlative or comparison is used without a clear expression of the direction or magnitude. I recommend articulating everything as concretely as possible.

I think they could lay out management goals and uncertainties and problems in the introduction so that it isn't such a shock in the conclusion. Discuss the management practices more in depth earlier on. And in the site description, you need to clearly describe the land cover as that is the most important part of this analysis. Over all I think there is a conflict between this being a "site specific" paper and a "method" paper that you need to handle more directly.

Specific Comments Note: I did not perform a detailed proof reading. But there are numerous syntax and tense problems that should be controlled. I'll put some here that stood out. For most of these, you do not need to reply to this line by line. Undoubtedly there are more or other ways that sentences could be reconfigured to solve the problem.

Page 1 L20 - no comma L22 - one mm is enough L24 - highly (also first sentence of discussion). What do you mean by this? Well constrained? What not just delete this word. L27 altered, complicated L28 remained L29 was L30 of rainfall L31 levels of water stress "Effectively use pulsed rainfall" => unclear perhaps you mean efficiently used sporadic rainfall??

Page 2 L2-4: These sentences seem identical to those in conclusions, can you say this more generally to launch paper? This whole first paragraph should really situate the research in the global problematic at the largest scale whereas here I feel shoved into a fairly narrow location, site, and quantitative description that seems drawn from a site description and conclusions. Perhaps frame it into the management needs of this ecosystem and why knowledge of transpiration would improve management? L1 an -> the (reference this first sentence?, not to prove your point but to guide the interested reader) Delete " in the form" add: grazing "land & fodder" in South Africa L2 change second half of sentence to ... is reevaporated in the form of .... Transpiration [ what does it mean to consume precipitation, really? ] L3: magnitudes The T component evaporates from leaf stomata of a ... L5 > 1 week = maybe for both extended and short (a few lines lower) just say periods lasting more than a week / sub-daily time scales

[Figure]

L12 is there a weird font of your references? L14 at time scales ranging between daily and seasonal L15 partition (delete -ing) L16 can you delete "display" ? L17 don't you talk about another species later? Also, this is a bit early to mention the study site. Please organize better. A species list would really help and some analysis of the relevant patch area. L22 in an hot and open ... but what do you mean by open here? Be more precise L 24 C4 draws uses soil water intensively and quickly However, to to their low water storage capacity, severe ..their LAD L25 Our objective is therefor... L26 environmental variables??? What are you referring to specifically and why? Do you mean environmental seasonality as measured by the following variables : It seems like your objective is more to compare methods than to partition. L27 to partition ET L26-28 this isn't so clear. Can you write about pros and cons / assumptions

Page 3 L2 - write equation in words followed by equation. It is really hard to read like that. Same for other places i.e. all the E(=ET-T) in line 5 and the whole following paragraph. L7 In this paper, three different methods that establish the relationship between ... L9 you talk about bins a lot. I'm not sure what you mean. Can you clearly define bin somewhere ? However some of this paragraph seems very technical for the intro. I would move some to methods. L13 I think you can remove "based on a recent review", the reference is enough L15-18 - this is too specific for the intro.

L29 1300 +/- 300 head of cattle according to year.

Site Description - how was station positioned in relation to farm? Protected land? Was the area around the station protected? A map and picture would help? What is the footprint of station?

PAge 4 L1 - specify the soil texture according to depth. L2 - change"water table depth" to not repeat - say water at 30 m below surface

L9 Introduce variables with something like In addition to other variables.... And then just focus on the one you use., be more precise "meteorological variables", at what time step, how many points, how chosen and organized in relation to land surface,

topography, etc.

L12 what direction does your EC station point? How does that compare with the dom-
inate rain. L17 how do you convert 2 way radiation to PAR? What's the foot print and
the land cover of its field of vision?

L20 two separate profiles L21 a single average soil moisture. This paragraph : where
in relation to other measurements - this would be good on a map L27 you say "mea-
surement" three times in one sentence - there must be a better way

IN GENERAL: how did you determine soil texture? Calibrate measurements? Validate
measurements?

Page 6 Section 2.5 - deja vu from intro - it makes sense to present it in detail here,
maybe you can talk about it more generally in intro . Page 7 First 5 lines - remember
to put goals in the intro and methods here. Also, in the intro you could compare the
methods in a more symbolic way and then here in the methods in a more technical
way. For example, a table that shows the variables they require, the output, some
examples of how and where they've been used (references), would really help the
reader conceptualize these 3 methods. L8 - "in the fitting" replace with "the process or
step of fitting the .... " for example

L21 sampled => selected L22 searched => identified L23 in 5 hydro years, there were
... .(comma and tense) L24 - write out CV and say what it means i.e. April has the most
variation of __ (cv = )

Page 8 Do you use mean storm frequency? Limit methods to things you use.

Page 9 Line 7 - the variation => that Table 1 legend. - it is plural stations? do you
present these stations ? That could go on a map What is in parenthesis and what is
the +/- ? And why not put the SD (last line) in the box with the mean like the others. ?
This table might be able to present graphically more succinctly.

A figure comparing measured rainfall between station by event would be very helpful

to interpret the accuracy of your measurement.

Table 2 - this could also be a figure i.e. with little creativity, tables 1 and 2 could be put on figure 1. "Having a late start" => starting late. This figure just looks so simple and unappealing, I really think you could make it more interesting and sophisticated with a little creativity

Fig 2 - make the 2011 dashed if you aren't considering it.

Page 13 - Figure 3 - my version is blurry. L6 - Define bin better somewhere!

m3m-3 ? Is that really the best way to write units? I think you should come up with another name for the T=ET line.

Figure 4. Relationship ! Should 2015 be included in the regression ?

Figure 5. Relationship !

Figure 6. It seems like this figure should come second, as it has to do with how you figured out T. It also could use some colors and creativity.

A scatter plot of the different methods would help highlight your comparison.

Tables 3 and 4 could be integrated into the other tables.

In general I think the figures could be streamlined a bit. Based on the objectives, I would say, present a) an over view of the data highlighting the seasonality, what was measured vs. what was calculated b) a plot with both magnitudes of evaporation and transpiration, comparing calculation methods, perhaps dry vs wet season +> P, E, T c) a large matrix of plots comparing environmental seasonality indicators (that you define as) with WB components d) remote sensing indicators compared with above

Figures that are really just to help understand methods can go in supplement.

Page 22 Line 3 - "highly" ??? Well? L5 - "aided" ? Augmented? L8 - this is a different tree than you previously said was important. Is this really the place for this deep

discussion? I really like the examination of specific trees and species but I think more work has to be done to tie it into this paper. L15 - choose slash or parenthesis P23 L25 add "a" before cultivated L29 - Beninese is the adjective, but is this really the only relevant research? This is a very different climate-vegetation zone from yours. It is not an open savanna. There is other research out there. P25 - L19-20 "Mediterranean" is it capital? I think this example (as with the previous comment) could be better integrated. If you compare climate zones, you need to articulate what you expect between these two climate zones. It is hard to make these comparisons.

L25 - what do you mean by conservative? L26 - are you clear about density and crops ?

P26L2 - do you mean consistent instead of conservative? L3 - conservative again? L6 - limitation - shortages?

L20 "farmers at the ranch" = ranchers

---

## Referee Comment (RC2) · Anonymous Referee #2 · 19 Jun 2020

The authors study the effect of rainfall amount and timing on transpiration in an area (South Africa) where not much knowledge is known on this topic. Therefore, topic-wise I very much welcome this study. However, the way the study is done, how it is presented, and how it's discussed needs major improvement.

Major points of attention:

1) The authors study ET-partitioning; however, they seem to only consider transpiration and soil evaporation. What about interception? Also in Savanna-areas interception can play a big role (15-30%). See for example the work of:

- Bulcock, H. H. and Jewitt, G. P. W.: Field data collection and analysis of canopy and litter interception in commercial forest plantations in the KwaZulu-Natal Midlands,

[Figure]

South Africa, Hydrol. Earth Syst. Sci., 16, 3717–3728, https://doi.org/10.5194/hess-16-3717-2012, 2012.

- Tsiko, C.T., Makurira, H., Gerrits, A.M.J., and Savenije, H.H.G. (2012): Measuring forest floor and canopy interception in a savannah ecosystem, Physics and Chemistry of the Earth, Vol. 47-48, 122-127

I think the authors should include interception in their analysis if possible and/or clear discuss this limitation and influence on the found results

2) I am puzzled by Table 1.

- How is it possible that ET>P on an annual time scale?!?! This can only be the case when irrigation is applied. However, no information is given on this. And in case irrigation is used, this should be added to P. Maybe I misunderstand something, but I think it is highly surprising that the authors do not explain this. They only state that P<ET (P9L5). Where is this water coming from? On an annual basis ET<=P, when you assume no discharge and storage change. Or are the ET values wrong due to the non-closure of the eddy covariance? Please explain this, quantify it and adjust your results (e.g., in case of an EBC-problem distribute the non-closure to H and LE based on the bowen ratio.

- The entire study build upon this ET that is larger then P, so I think the meaning of the T/ET-ratios are not meaningful.

- Furthermore, I do not understand the values of P. In the caption it is written that these are the values of the cite and that the values between brackets are from a nearby site. However, in the text (P9L1) the author say that the nearby site has higher rainfall amount, which isn't shown in the table. Please clarify, check and possible correct.

3) Presentation: Honestly, I am not an expert in the applied methods to estimate transpiration; however, I am familiar with ET-partitioning. Having said this, I think the authors can do a better job in explaining their used methods, so it's better understandable

for a broader audience.

Minor:

- abstract: It's good practice to add the knowledge gap in your abstract (i.e. "missing knowledge on carbon uptake")

- units: throughout the manuscript the units are not correct. When speaking about annual rainfall/ ET, T, E the unit is mm/YEAR and not just mm. Please correct

- style: the HESS style states the parameters (ET, P, T, etc) should be in italic.

- P2L6-7: I do not understand this sentence.

- P5L3: What is WPL?

- P5 eq 1: add units to all symbols

- P6L8: I am not getting this. $E_{u,k}$ should be unitless if I see eq 2...

- P6L9: Explain MDS, LE, and add that sigma^2 is variance

- P7L4-6: what is the naming of this method uWUE or uWUEp. Please be consistent

- fig 2: y-axis should be labeled: Accumulated monthly transpiration [mm]

- fig 7: what is the reasoning for using power-functions and not e.g., linear ones?

- fig 9: unit of y-axis is mm/day

---

## Author Comment (AC1) · 17 Jul 2020

We thank the reviewers for the positive reception of the manuscript, as well as all their comments and helpful suggestions. We have explicitly reported the nighttime ET results and discussed its influence on ET uncertainty. The language and structure were also revised in accordance with their suggestions.

**Anonymous Reviewer #1**

Thank you for the article about the effect of rainfall amount and timing on annual transpiration in the grazed savanna grassland.

This is an extremely interesting paper on a very important subject in an overall datascarce region. I believe that overall the methods and scientific work is sound, however I do think that the paper would benefit from some editing and streamlining. Often sentences are long and cumbersome and the presentation seems redundant and not distinct. Thus most of my feedback is on presentation quality and not scientific quality or significance. I believe it does present a substantial contribution to scientific progress within the scope of Hydrology and Earth System Sciences in terms new concepts, ideas, methods, and data. I believe the scientific approach and applied methods to be valid. I believe that the results are discussed in an appropriate and balanced way.

They consider some related work but have some pretty large omissions. They only invoke one other, non-directly-related, study using eddy-covariance in Africa. This study was in Benin, in a very different climate zone. Meanwhile, if you include the whole African continent, there are have been other studies using eddy-covariance, many in more similar climatic areas. It would be interesting to compare your results with those of more similar studies, for example see the work of Marc Parlange's group in Burkina Faso, or group by the French CIRAD organization and associated researchers throughout francophone west Africa, or Kelly Caylor's work in East Africa, or some older studies in Niger and Nigeria.

Thank you for this suggestion. The focus of the analysis is on the transpiration and we now discuss transpiration estimates from different studies. In addition, we have now added comparison to the Ramier et al (2009) annual ET measurements.

One very minor, but very significant change in presentation would be to format paragraphs as distinct. I found it very hard to read because the appearance is of one single continual paragraph. Perhaps this is the flaw with the HESS template, but a compromise should be found!

We are using the HESS template. We now divide paragraphs into shorter ones.

Additionally, although the information that I would want from a picture of the site and a map is present (geographical coordinates, density of trees), I think it would be a tremendous boost to the ability of the reader to visualize if at least one figure was added including a map, a picture of the measurement station, a picture of the land cover (topography) around the station, and perhaps a sketch (to scale) of the different components of evapotranspiration (i.e. from C3 grasses, C4 grasses, woody-vegetation, ground water depth, soil layers, and any surface water etc. ).

We have overlaid a 2D footprint analysis on the site map so as to indicate the tree and land cover in proximity of the site together with a picture of the station as requested (Fig. 1).

I rate this paper Good (2) because I don't think the results and conclusions are currently presented in a clear, concise, and well-structured way. Some figures could be integrated with each other and of higher quality. The English is correct, though I think some editing (fewer words) would make it more readable.

We restructured and streamlined the discussion of several sections, including the soil physics evaporation check. This is now included as supplementary material because it offers an indirect (but independent) check on evaporation estimates. Figure 5 and 7 were combined, and only Table 1 remains in the results. Others figures are integrated with existing ones or moved to supplementary material.

The paper addresses (scientific questions):

Objective: to partition measured ET Quantify effect of seasonality on grass activity (this is very vague, and it is not clear why this is important).

Transpiration relates to productivity of these ecosystems as well as grazing. We have added the following sentence:
*"These longer time scales are of interest to valuation of ecosystem productivity and their services when assessing climatic shifts (Godde et al. 2020)."*

How ET and its components vary according to mean annual P and vegetation change in a grazed savanna grassland ecosystem 3 study objectives: 1) quantifying the variation in P, ET, and estimated T based on met measurements at this new long term monitoring site 2) identifying the main drivers of the annual, seasonal, and monthly variation in the water balance components 3) identifying remote sensing variables that explain variations in T/ET and transpiration.

Later the goal is to select the method most applicable to water limited ecosystems.

I think the paper would be strengthened if they adhered more to these goals and used them to structure the results, discussion, and even conclusions. I will try to reproduce the calculation with a similar data set, and I currently believe that you present ample information even if I have not yet tried. The title clearly placed annual transpiration as the goal. It does not invoke the three methods. The title is acceptable but you could make it more exciting or intriguing. Overall, I think they should write out abbreviations and equations more often in full text. Should any parts of the paper (text, formulae, figures, tables) be clarified, reduced, combined, or eliminated?

Thank you for this suggestion. We streamlined the manuscript by moving the soil physics check to the supplement. The beginning of the results section introduces the blue-print of how the results addresses the goals of the study.

I found some sentences to be repeated more or less word for word in multiple parts of the text. This is completely unacceptable. Please respect your reader who reads all the way through and reword. Each sentence should bring new information. i.e. the first sentence of the introduction is the same as the short summary. For the short summary, I recommend saying some unknown problem and generalizing more. In general, sentences are jargon-y & convoluted. Some frames that could be said differently are repeated. I.e. partitioning? You could alternate it with divided, split, etc and talk about it more concretely. There are places where the article is missing. There are numerous places where a superlative or comparison is used without a clear expression of the direction or magnitude. I recommend articulating everything as concretely as possible.

We revised the language and vary the expression appropriately in different sections. The comparisons have been adjusted.

I think they could lay out management goals and uncertainties and problems in the introduction so that it isn't such a shock in the conclusion. Discuss the management practices more in depth earlier on. And in the site description, you need to clearly describe the land cover as that is the most important part of this analysis. Overall I think there is a conflict between this being a "site specific" paper and a "method" paper that you need to handle more directly.

Thank you for this suggestion. We view the analysis here as 'necessary but not sufficient' for constructing and implementing best management practices. Specific recommendations about management practices of these type of grasslands is beyond the scope of this study. However, we do now state that the site is under heavy grazing each year that limits the grass height to 10 cm or less. The uncertainty in water balance is now presented in both – the results and discussion sections. The site description is now more detailed and includes a map of the land and tree cover.

Specific Comments Note: I did not perform a detailed proof reading. But there are numerous syntax and tense problems that should be controlled. I'll put some here that stood out. For most of these, you do not need to reply to this line by line. Undoubtedly there are more or other ways that sentences could be reconfigured to solve the problem.

Thanks, the syntax and tenses have been checked.

Page 1 L20 - no comma L22 - one mm is enough L24 - highly (also first sentence of discussion). What do you mean by this? Well constrained? What not just delete this word. L27 altered, complicated L28 remained L29 was L30 of rainfall L31 levels of water stress "Effectively use pulsed rainfall" => unclear perhaps you mean efficiently used sporadic rainfall??

Corrected. Removed highly. Changed to sporadic rainfall.

Page 2 L2-4: These sentences seem identical to those in conclusions, can you say this more generally to launch paper? This whole first paragraph should really situate the research in the global problematic at the largest scale whereas here I feel shoved into a fairly narrow location, site, and quantitative description that seems drawn from a site description and conclusions. Perhaps frame it into the management needs of this ecosystem and why knowledge of transpiration would improve management? L1 an -> the (reference this first sentence?, not to prove your point but to guide the interested reader) Delete " in the form" add: grazing "land & fodder" in South Africa L2 change second half of sentence to ... is reevaporated in the form of .... Transpiration [ what does it mean to consume precipitation, really? ] L3: magnitudes The T component evaporates from leaf stomata of a ... L5 > 1 week = maybe for both extended and short (a few lines lower) just say periods lasting more than a week / sub-daily time scales

Added reference and revised the first paragraph.

L12 is there a weird font of your references? L14 at time scales ranging between daily and seasonal L15 partition (delete -ing) L16 can you delete "display" ? L17 don't you talk about another species later? Also, this is a bit early to mention the study site. Please organize better. A species list would really help and some analysis of the relevant patch area. L22 in an hot and open ... but what do you

mean by open here? Be more precise L 24 C4 draws uses soil water intensively and quickly However, to to their low water storage capacity, severe ..their LAD L25 Our objective is therefor... L26 environmental variables??? What are you referring to specifically and why? Do you mean environmental seasonality as measured by the following variables : It seems like your objective is more to compare methods than to partition. L27 to partition ET L26-28 this isn't so clear. Can you write about pros and cons / assumptions

Corrected. Here the old name (Acacia) was mistakenly used. It is the same species. We have clarified the role of the method comparison.

Page 3 L2 - write equation in words followed by equation. It is really hard to read like that. Same for other places i.e. all the E(=ET-T) in line 5 and the whole following paragraph. L7 In this paper, three different methods that establish the relationship between ... L9 you talk about bins a lot. I'm not sure what you mean. Can you clearly define bin somewhere ? However some of this paragraph seems very technical for the intro. I would move some to methods. L13 I think you can remove "based on a recent review", the reference is enough L15-18 - this is too specific for the intro.

This section was moved to methods and the sentences were revised.

L29 1300 +/- 300 head of cattle according to year.

Corrected

Site Description - how was station positioned in relation to farm? Protected land? Was the area around the station protected? A map and picture would help? What is the footprint of station?

The housing is 300 m away from the measurement station located on a large commercial farm. The measurement station is inside a fence. We added a picture of the measurement site and a satellite map of the area with a 2D flux footprint estimated from daytime data (Fig. 1).

PAge 4 L1 - specify the soil texture according to depth. L2 - change"water table depth" to not repeat - say water at 30 m below surface

Soil texture represents surface soil but there are no changes in texture at least up to a depth of 1 m.

L9 Introduce variables with something like In addition to other variables.... And then just focus on the one you use., be more precise "meteorological variables", at what time step, how many points, how chosen and organized in relation to land surface, topography, etc.

All the variables are measured inside or next to the measurement station. The nearest patch of trees is about 15 m away from measurements. The meteorological measurements were sampled every minute and 15 min averages were recorded.

L12 what direction does your EC station point? How does that compare with the dominate rain. L17 how do you convert 2 way radiation to PAR? What's the foot print and the land cover of its field of vision?

The EC sensors are pointing towards the north. The mean wind direction during rainfall was 124°. The PAR is separately measured by Kipp & Zonen PAR-lite sensors. All radiation sensors are positioned at 3 m above the ground with field of view at the grass field near the measurement station (this is added in the revised manuscript).

L20 two separate profiles L21 a single average soil moisture. This paragraph : where in relation to other measurements - this would be good on a map L27 you say "measurement" three times in one sentence - there must be a better way

Corrected and revised the sentence.

IN GENERAL: how did you determine soil texture? Calibrate measurements? Validate measurements?

The detailed soil sampling results are reported in a previous publication cited here (Räsänen et al., 2017). Also, all the details about the calibration of the eddy covariance system are provided in that publication. The soil moisture sensors were calibrated for the soil according to the manufacturer manual.

*The gas analyser was calibrated every month with a high-accuracy $CO_2$ span gas (378 ppm verified by the Cape Point GAW station), and Afrox instrument grade synthetic air with $CO_2$ < 0.5 ppm was continuously used as a reference gas.*

Page 6 Section 2.5 - deja vu from intro - it makes sense to present it in detail here, maybe you can talk about it more generally in intro . Page 7 First 5 lines - remember to put goals in the intro and methods here. Also, in the intro you could compare the methods in a more symbolic way and then here in the methods in a more technical way. For example, a table that shows the variables they require, the output, some examples of how and where they've been used (references), would really help the reader conceptualize these 3 methods. L8 - "in the fitting" replace with "the process or step of fitting the .... " for example

The method details in the introduction were moved to Section 2.5. We added a table that shows input variables, references and an explanation how monthly T/ET is calculated for each method.

L21 sampled => selected L22 searched => identified L23 in 5 hydro years, there were ... .(comma and tense) L24 - write out CV and say what it means i.e. April has the most variation of __ (cv = )

Corrected

Page 8 Do you use mean storm frequency? Limit methods to things you use.

Yes, it is presented in table 2 and now added to the figure 1.

Page 9 Line 7 - the variation => that Table 1 legend. - it is plural stations? do you present these stations ? That could go on a map What is in parenthesis and what is the +/- ? And why not put the SD (last line) in the box with the mean like the others. ? This table might be able to present graphically more succinctly.

The table caption was revised as follows:

*Table 1. Annual sum of water balance components for each hydrological year (September to August). The annual P at the measurement site is followed by the annual P at the SAWS Potchefstroom station in parentheses for comparison. The total uncertainty (Eq. 3) is indicated for ET after $\pm$ sign. $ET_N$ is the annual nighttime evapotranspiration. The PET, determined from Eq. 1 is also shown. The transpiration and evaporation are calculated from monthly T/ET estimates. The EBC-slope stands for the slope of the energy balance closure with ordinate defined by measured $R_n$-G and abscissa defined by the sum of the measured latent and sensible heat fluxes.*

A figure comparing measured rainfall between station by event would be very helpful to interpret the accuracy of your measurement.

The annual rainfall between the measurement site and SAWS station is not significantly correlated and thus the event scale analysis is not possible. The SAWS data has many gaps in 2010 but not during the other years. The fact that the SAWS rainfall is 139 mm yr$^{-1}$ higher than the measurement site rainfall in 2011 and it is lower than the site rainfall during all the other years supports the fact that the rainfall at the site may have been underestimated in 2011.

Table 2 - this could also be a figure i.e. with little creativity, tables 1 and 2 could be put on figure 1. "Having a late start" => starting late. This figure just looks so simple and unappealing, I really think you could make it more interesting and sophisticated with a little creativity

Table 1 was kept in the results and annual nighttime ET was added to the table. The precipitation statistics were added to Figure 1 and Table 2, which was moved to the supplement.

Fig 2 - make the 2011 dashed if you aren't considering it.

The year 2011 is indicated with a star and not considered in the linear regression due to uncertain precipitation.

Page 13 - Figure 3 - my version is blurry. L6 - Define bin better somewhere!

The figure resolution was changed and the bins were defined also here in the caption.

m3m-3 ? Is that really the best way to write units? I think you should come up with another name for the T=ET line.

According to HESS guidelines units should be written exponentially. The T=ET line is a slightly awkward expression, but it emphasizes the method assumption.

Figure 4. Relationship ! Should 2015 be included in the regression ?

Corrected. The fit line was slightly extended to indicate that 2015 is included in the fit.

Figure 5. Relationship !

Corrected

Figure 6. It seems like this figure should come second, as it has to do with how you figured out T. It also could use some colors and creativity.

Figure 6 is positioned at the comparison.

A scatter plot of the different methods would help highlight your comparison.

We prefer the time series plots that reveal the strong seasonality at the site and support the method check. A scatter plot would add statistical measures of the differences, which are not the focus here.

Tables 3 and 4 could be integrated into the other tables.

The tables were moved to the supplement.

In general I think the figures could be streamlined a bit. Based on the objectives, I would say, present a) an over view of the data highlighting the seasonality, what was measured vs. what was calculated b) a plot with both magnitudes of evaporation and transpiration, comparing calculation methods, perhaps dry vs wet season +> P, E, T c) a large matrix of plots comparing environmental seasonality indicators (that you define as) with WB components d) remote sensing indicators compared with above

Figure 5 and 7 were combined. Table 2 precipitation statistics were added to Figure 1. The text in hydrological years section was streamlined. The structure of the results is briefly explained at the beginning and it corresponds to the aim of the study.

Figures that are really just to help understand methods can go in supplement.

Page 22 Line 3 - "highly" ??? Well? L5 - "aided" ? Augmented? L8 - this is a different tree than you previously said was important. Is this really the place for this deep discussion? I really like the examination of specific trees and species but I think more work has to be done to tie it into this paper.

It is the same tree species. The old name is Acacia. The dry season transpiration, as well as increase of early season T/ET with tree green-up are all results related to this specific dominant tree at the site.

L15 - choose slash or parenthesis P23 L25 add "a" before cultivated L29 - Beninese is the adjective, but is this really the only relevant research? This is a very different climate-vegetation zone from yours. It is not an open savanna. There is other research out there.

There are definitely more studies on ET and we have added the Ramier et al. (2009) study from a similar climate to the ET comparison. If there are other studies reporting measured annual ET from savannas with deep rooted trees, we can add a comparison. However, the analysis is focused on transpiration that is compared to other studies. We agree that the Beninese site is not similar but our comparison focuses on the dry season when sites are more similar.

 P25 - L19-20 "Mediterranean" is it capital? I think this example (as with the previous comment) could be better integrated. If you compare climate zones, you need to articulate what you expect between these two climate zones. It is hard to make these comparisons.

In this comparison we have pointed out the important difference in soils that is the most likely reason for the difference in T/ET. We have not found any other relevant study reporting monthly T/ET results.

L25 - what do you mean by conservative? L26 - are you clear about density and crops ?

Changed to "nearly constant". The 15 % tree cover means "few trees".

P26L2 - do you mean consistent instead of conservative? L3 - conservative again? L6 - limitation - shortages?

Nearly constant. Corrected.

L20 "farmers at the ranch" = ranchers

Corrected

References

Ramier, D., Boulain, N., Cappelaere, B., Timouk, F., Rabanit, M., Lloyd, C. R., Boubkraoui, S., Métayer, F., Descroix, L. and Wawrzyniak, V.: Towards an understanding of coupled physical and biological processes in the cultivated Sahel – 1. Energy and water, Journal of Hydrology, 375(1–2), 204–216, doi:10.1016/j.jhydrol.2008.12.002, 2009

Räsänen, M., Aurela, M., Vakkari, V., Beukes, J. P., Tuovinen, J.-P., Van Zyl, P. G., Josipovic, M., Venter, A. D., Jaars, K., Siebert, S. J., Laurila, T., Rinne, J. and Laakso, L.: Carbon balance of a grazed savanna grassland ecosystem in South Africa, Biogeosciences, 14(5), 1039–1054, doi:10.5194/bg-14-1039-2017, 2017.

---

## Author Comment (AC2) · 17 Jul 2020

We thank the reviewers for the positive reception of the manuscript, as well as all their comments and helpful suggestions. We have explicitly reported the nighttime ET results and discussed its influence on ET uncertainty. The language and structure were also revised in accordance with their suggestions.

**Anonymous Reviewer #2**

The authors study the effect of rainfall amount and timing on transpiration in an area (South Africa) where not much knowledge is known on this topic. Therefore, topic-wise I very much welcome this study. However, the way the study is done, how it is presented, and how it's discussed needs major improvement.

Major points of attention:

1) The authors study ET-partitioning; however, they seem to only consider transpiration and soil evaporation. What about interception? Also in Savanna-areas interception can play a big role (15-30%). See for example the work of:

- Bulcock, H. H. and Jewitt, G. P. W.: Field data collection and analysis of canopy and litter interception in commercial forest plantations in the KwaZulu-Natal Midlands, South Africa, Hydrol. Earth Syst. Sci., 16, 3717–3728, https://doi.org/10.5194/hess- 16-3717-2012, 2012.

- Tsiko, C.T., Makurira, H., Gerrits, A.M.J., and Savenije, H.H.G. (2012): Measuring forest floor and canopy interception in a savannah ecosystem, Physics and Chemistry of the Earth, Vol. 47-48, 122-127

I think the authors should include interception in their analysis if possible and/or clear discuss this limitation and influence on the found results

Thank you for this comment. We agree tha interception is important, especially with higher tree cover. In the Berkelhammer method we exclude rainy days from the calculation of half-hour T/ET values and thus those values do not include any interception. Although the eddy covariance system could be measuring interception in ET, this cannot be a large part of measured ET because most of the rainfall happens in the afternoon and evening. The mean maximum grass height was 10 cm for a one-year vegetation sampling period reported in a previous publication. Most of the grass is lower than this height due to the heavy grazing. If we assume 1 mm storage capacity per rainfall event for the grass, then 50 rainfall events per year will result to 50 mm interception (9% of annual rainfall). This assumes that all stored water is evaporated, which may not be true during nighttime conditions.

2) I am puzzled by Table 1.

- How is it possible that ET>P on an annual time scale?!?! This can only be the case when irrigation is applied. However, no information is given on this. And in case ir- rigation is used, this should

be added to P. Maybe I misunderstand something, but I think it is highly surprising that the authors do not explain this. They only state that P<ET (P9L5). Where is this water coming from? On an annual basis ET<=P, when you assume no discharge and storage change. Or are the ET values wrong due to the non-closure of the eddy covariance? Please explain this, quantify it and adjust your results (e.g., in case of an EBC-problem distribute the non-closure to H and LE based on the bowen ratio.

We have restructured the Results section to address this issue. There is no irrigation at the site. One reason is the underestimation of rainfall due to (i) well-known issues with tipping bucket gauges and (ii) scale mismatch between rainfall and ET measurements. In the revision, we show explicitly that the gap-filled nighttime ET varies from 58 to 85 mm yr$^{-1}$. If the nighttime ET was assumed to be zero, then the ET/P would be below 1.0 for four years.

The use of the Bowen ratio method to force closure on the energy balance would actually add uncertainty. This type of forced closure requires us to assume that the two heat fluxes are underestimated by the same amount, and thus the Bowen ratio of the eddy covariance measured fluxes and Rn-G are correctly. The underestimation of the heat fluxes is often attributed to large-scale eddies (low frequency contribution to the co-spectrum of the two fluxes), which are impacted by entrainment processes and variability in sources and sinks at the land surface. Both of these are not similar for heat and water vapor. It is difficult to state how much of non-closure should be attributed to latent heat flux only. Also, it is worth noting that the net radiation measurement (Rn) is measuring only grass.

- The entire study build upon this ET that is larger then P, so I think the meaning of the T/ET-ratios are not meaningful.

Our conjecture is that the underestimation is on precipitation side and the spatial scale mismatch between ET and P. There is also gap-filling of nighttime ET that is an issue when dealing with annual hydrologic balance (less of an issue when portioning ET to E and T for a given averaging period).

- Furthermore, I do not understand the values of P. In the caption it is written that these are the values of the cite and that the values between brackets are from a nearby site. However, in the text (P9L1) the author say that the nearby site has higher rainfall amount, which isn't shown in the table. Please clarify, check and possible correct.

We have clarified the meaning of the rainfall measurements from the SAWS station. The fact that the SAWS rainfall is 139 mm yr$^{-1}$ higher than the measurement site rainfall in 2011 and lower than the site rainfall during all the other years, supports the fact that the rainfall at the site was underestimated in 2011 due to poor performance of the tipping bucket sensor during heavy rainfall.

3) Presentation: Honestly, I am not an expert in the applied methods to estimate transpiration; however, I am familiar with ET-partitioning. Having said this, I think the authors can do a better job in explaining their used methods, so it's better understandable for a broader audience.

We have added a table explaining the methods (assumptions and input) and revised this section accordingly.

Minor:

- abstract: It's good practice to add the knowledge gap in your abstract (i.e. "missing knowledge on carbon uptake")

Stated as the role of precipitation variability on ET, transpiration and evaporation.

- units: throughout the manuscript the units are not correct. When speaking about annual rainfall/ ET, T, E the unit is mm/YEAR and not just mm. Please correct

Corrected

- style: the HESS style states the parameters (ET, P, T, etc) should be in italic.

Corrected

- P2L6-7: I do not understand this sentence.

The sentence was revised. The evaporation from soils can persist over extended periods lasting more than a week.

- P5L3: What is WPL?

The standard Webb-Pearman-Leuning (WPL) density correction.

- P5 eq 1: add units to all symbols

Corrected

- P6L8: I am not getting this. E_u,k should be unitless if I see eq 2…

This ratio is multiplied by annual ET that was missing from the equation

- P6L9: Explain MDS, LE, and add that sigma^2 is variance

It's Marginal Distribution Sampling that was defined in the gap-fill section. The section was revised.

- P7L4-6: what is the naming of this method uWUE or uWUEp. Please be consistent

This sentence was revised. The name of the method is Uwue, while uWUEp is the name for the slope of the fitted T=ET line.

- fig 2: y-axis should be labeled: Accumulated monthly transpiration [mm]

Corrected

- fig 7: what is the reasoning for using power-functions and not e.g., linear ones?

The points are fitted using linear regression.

- fig 9: unit of y-axis is mm/day

Corrected